

# In-Flight Calibration of SCIAMACHY's Polarization Sensitivity

Patricia Liebing[1,3], Matthijs Krijger[2,4], Ralph Snel[2,5], Klaus Bramstedt[1], Stefan Noël[1],
Heinrich Bovensmann[1], and John P. Burrows[1]

[1]Institute of Environmental Physics (IUP), University of Bremen, Germany
[2]SRON Netherlands Institute for Space Research, Sorbonnelaan 2, 3584 CA Utrecht, the Netherlands
[3]Now at Leiden Observatory, 2300 RA Leiden, the Netherlands
[4]Now at Earth Space Solutions, Utrecht, the Netherlands
[5]Currently at TNO, Stieltjesweg 1, 2628 CK Delft, the Netherlands

*Correspondence to:* Patricia Liebing (Patricia.Liebing@iup.physik.uni-bremen.de)

**Abstract.** This paper describes the in-flight calibration of the polarization response of the SCIAMACHY Polarization Measurement Devices and a selected region of its science channels. With the lack of polarized calibration sources it is not possible to obtain such a calibration from dedicated calibration measurements. Instead, the Earth shine itself, together with a simplified Radiative Transfer Model, is used to derive time- and measurement configuration dependent polarization sensitivities.

The results are compared to an instrument model that describes the degradation of the instrument as a result of a slow build up of contaminant layers on its elevation and azimuth scan mirrors. This comparison reveals significant differences between the model prediction and the data, suggesting an unforeseen change between on-ground and in-flight in at least one of the polarization sensitive components of the optical bench. The possibility of mechanisms other than scan mirror contamination contributing to the degradation of the instrument will be discussed. The data are consistent with a polarization phase shift

occurring in the beam split prism used to divert the light coming from the telescope to the different channels and Polarization Measurement Devices. The extension of the instrument degradation model with a linear retarder enables the determination of the relevant parameters to describe this phase shift, and ultimately results in a significant improvement of the polarization measurements as well as the polarization response correction of measured radiances.

## 1 Introduction

SCIAMACHY (Bovensmann et al., 1999) was a passive spectrometer on board ESA's ENVISAT platform, operational between 2002 and 2012. Spectra of the radiance upwelling at the top of the atmosphere in nadir limb and occultation geometries, and extraterrestrial solar irradiance were measured by SCIAMACHY. The uncalibrated level 0 data are transformed by applying the instrument instrument response functions, obtained during pre-flight and in-flight calibration, to the radiance, irradiance and reflectance. Measurements between 214 and 1750 nm were contiguously recorded in 6 channels at wavelength dependent

spectral resolutions between 0.2 and 1.5 nm. In addition, there were 2 channels with bands around 2.0 microns and 2.3 microns and spectral resolutions around 0.2 nm and 7 polarization monitoring devices, PMDs. The latter were broad band detectors at selected wavelengths. Combined with the spectral channels they were used to determine the polarization of the upwelling radiation. The resulting radiance, irradiance and reflectance data products were used to derive column amounts and concentrations





of atmospheric trace gases and aerosol and cloud data products. The accuracy of these data products depends critically on that of the measured spectra, i.e., the absolute magnitude of the radiometric signals and their spectral shape. Therefore it is vital to obtain a highly accurate characterization of the instrument response functions to unpolarized and polarized light, both from on-ground measurements and in-flight. SCIAMACHY is equipped with scan mirrors to reflect the light from nadir or limb

line-of-sights into the telescope and then further towards grating spectrometers and photodiode detectors. The degradation of the scan mirrors can be modeled with a thin, slowly growing layer of contaminant on top of their oxidized aluminum surfaces. In-flight calibration measurements for SCIAMACHY were carried out regularly with unpolarized sources, mainly the sun, and for different scan mirror configurations and positions. These data have been analyzed with the so called scan mirror model which, by applying the Fresnel equations for layered surfaces to these specific measurement configurations, determined the

refractive indices and thicknesses of the contaminant on each of the two scan mirrors (Krijger et al., 2014). Indeed, a major part of the observed instrumental throughput degradation in the UV, and its scan angle dependence, can be explained by the contamination of the scan mirrors. The remaining part is attributed to the degradation of the optical bench module (OBM), which comprises all elements of the optical train behind the instrument slit.

The spectrometer grating, but also the scan mirrors and other optical components along the light path are polarization

sensitive, meaning that their optical properties depend partially on the polarization of the incident light. The polarization sensitivity of the instrument as a whole has been measured on-ground (Gottwald and Bovensmann, 2011), and found to be substantial, in the order of 20% or more, over large wavelength ranges. The measured polarization values, obtained from the combination of PMD and science channel measurements, can be used to correct the signal of the SCIAMACHY science channels. For this correction to be effective, both the polarization and the polarization response have to be known with sufficient

accuracy. Measured polarization and reflectance values in the visible and near infrared range have been compared to POLDER data and found to be within the expected range for selected geometries in nadir (Tilstra and Stammes, 2007, 2010). Limb polarization data on the other hand differed systematically from Radiative Transfer Model (RTM) results (Liebing et al., 2013), an effect which is largest in the UV and decreases towards longer wavelengths. Of importance is the observation that, even though the data show a significant drift over time, a difference between in in-flight and pre-flight measurements is

already present early in the mission. These findings imply that there were either unidentified systematic errors in the on-ground calibration or perhaps a sudden change of polarization calibration parameters shortly after launch, followed by an in-flight degradation of polarization sensitive components. Due to the variation of the observation geometry – and therefore of the polarization – with latitude and season, errors in the polarization determination or correction may lead to latitudinally and seasonally dependent systematic errors in the derived atmospheric trace gas concentrations. Moreover, longterm degradation

effects may consequently lead to drifts in the data if not appropriately accounted for.

Recently, the on-ground calibration data have been reanalyzed, making use of the scan mirror model for SCIAMACHY (Krijger et al., 2014). The application of this model for uncontaminated mirrors during the on-ground calibration campaign enables the separation of the radiance and polarization sensitivities of the scan mirrors from that of the OBM. The knowledge of the OBM polarization response can then be combined with that on the mirror contamination in space derived from the in-





flight calibration measurements. The transformation to nadir and limb measurement configurations yields a prediction of the time dependent behavior of in-flight polarization sensitivities for limb and nadir observations.

The aim of this paper is to devise a method for the direct in-flight calibration of the polarization response which enables a validation of the scan mirror model and its derived parameters as well as a correction for the suspected on-ground to in-flight
change. Furthermore, an investigation of this on-ground to in-flight change leads to its most likely culprit: a polarization phase shift induced in the beam split prism that is used to separate the almost completely polarized beams for the PMDs from the partially polarized beams for the science detectors. Such a phase shift can be modeled with an extension to the scan mirror model that includes a linear retarder and thus modifies the OBM polarization sensitivities as measured on-ground.

A polarized calibration source is necessary in order to derive the in-flight polarization sensitivities. As the sun and on-board
calibration sources are unpolarized, the only available polarized calibration source is the Earth shine itself. By exploiting the relationship between depolarization and intensity at fixed wavelength and viewing geometry, data points with "known" polarization can be selected and sampled in order to obtain the polarization sensitivities. The information available in the data permits this to be done only for the PMDs, and for a particular feature in SCIAMACHY's science channel at around 350 nm. The statistical nature of this approach mitigates errors arising from the underlying assumptions to a certain degree, the
remaining errors are to be estimated from the systematic variation of those assumptions. The results obtained in the UV and, partially, VIS regions prove to be accurate enough to deduce from them the parameters of the instrumental phase shift and its possible cause. This investigation may also be relevant in view of other instruments whose design follows a similar principle, such as the GOME series on the ERS-2 and MetOp satellites.

The structure of this article is as follows: In Sect. 2, first the polarization formalism used here is introduced, including the
Mueller matrix approach and the description of polarization coordinate frame. Next, the SCIAMACHY polarization measurements are explained in detail. Section 3 explains the calibration strategy and presents the results. The interpretation of the results in terms of an on-ground to in-flight change of the OBM, and possibly its further degradation is presented in Sect. 4. This section also contains a discussion on implications regarding the status of the instrument before and after launch, and on the applicability of the scan mirror model. Conclusions are presented in Sect. 5.

## 2   Polarization Formalism and Measurements

Throughout this paper, we employ the Mueller Matrix formalism to describe the instruments sensitivity to polarized light. The formalism, frame definitions and application to SCIAMACHY are presented in this section together with a description of the polarization measurement approach.

### 2.1   Mueller Matrix Formalism and Polarization Frame

Formally, the measured signal of any given polarization sensitive instrument is the first element of the Stokes vector resulting from the product of the incoming Stokes vector $\boldsymbol{I} = (I, Q, U, V)^T$ with the $4 \times 4$-Mueller matrix that describes its modification





by the optical elements inside the instrument:

$$\boldsymbol{S}_0 = [\mathbf{M} \cdot \boldsymbol{I}]_0 . \tag{1}$$

The Stokes vector components are combinations of the intensities $I$ with different polarization planes with respect to a given reference frame:

$$I = I_\| + I_\perp , \quad Q = I_\| - I_\perp , \quad U = I_{45°} - I_{-45°} , \quad V = I_R - I_L . \tag{2}$$

The linear polarization is defined by the $Q$ and $U$ components, $V$ is the circular polarization component. Often, the Stokes vector is given in relative terms, i.e., $\boldsymbol{I} = I(1, q, u, v)^T$, where the second to fourth components can take values between -1 and 1 and the total degree of polarization is defined as

$$p \equiv \sqrt{q^2 + u^2 + v^2} \leq 1. \tag{3}$$

The Stokes vector frame is defined by choosing the parallel ($\|$), axis. The perpendicular ($\perp$) axis is then naturally given, while for the $\pm 45°$ direction as well as the left- ($L$) and right- ($R$) handedness of the circular polarization a sense of orientation needs to be defined. Here, the sense of orientation is given by looking into the travel direction of the light, i.e., towards the instrument. A positive rotation is defined to be clock wise, i.e., left-handed.

The elements of the Mueller matrix can be interpreted in the following way: $M_{11}$ is the optical throughput for unpolarized light, i.e., $\boldsymbol{I} = (I, 0, 0, 0)^T$. The other diagonal elements $M_{ii}$ describe the transfer efficiency for a given input Stokes vector component ($Q$ for $i = 2$, $U$ for $i = 3$, $V$ for $i = 4$) to the output Stokes vector, while the non-diagonal elements stand for the cross talk between different Stokes vector components. The instrument Mueller matrix itself is a product of the Mueller matrices of its individual components. The Mueller matrix needs to be given in the same reference frame as the Stokes vector, if necessary, frames defined differently for individual components have to be transformed to match.

Since only the first component of the signal Stokes vector $\boldsymbol{S}_0 \equiv S$ is being measured, only the first row of the instrument Mueller matrix is needed for calibration of the detector signal:

$$S = I M_{11}(1, \mu_2, \mu_3, \mu_4)(1, q, u, v)^T, \tag{4}$$

where the $\mu_i = M_{1i}/M_{11}$ are the normalized Mueller matrix elements. Note that, if the end-to-end Mueller matrix is derived from a combination of individual matrices, all but the last one of those still have to be known completely.

For SCIAMACHY, the Stokes frame for the measured atmospheric polarization is defined with respect to the local meridional plane, the plane spanned by the line-of-sight and the local zenith. Parallel polarization, i.e. $q = +1$, is given when the polarization lies in this plane. Positive $45°$ polarization $u = +1$ is defined for the polarization plane which is rotated clock wise by $45°$ from the parallel direction, when looking along the line-of-sight into the instrument. For calibration purposes and within the scan mirror model, another frame definition is applied. The end-to-end Mueller matrix or Stokes vectors can easily be transformed between these different frames. In this paper we work entirely with the atmospheric polarization, and to avoid confusion, we choose to give the end-to-end Mueller vector in the atmospheric polarization frame.





## 2.2 The scan mirror model

The details of the scan mirror model and its application to SCIAMACHY are described in (Krijger et al., 2014). Essentially, the instrument is separated into a scan module consisting of one or several scan mirrors, and the optical bench module which comprises everything else, i.e.,

$$5 \quad S = I M_{11}^{OBM} \boldsymbol{\mu}^{OBM} \mathbf{M}(\alpha)(1, q, u, v)^T. \tag{5}$$

Here, $\boldsymbol{\mu}^{OBM}$ is the first row vector of the normalized OBM Mueller matrix, $M_{11}^{OBM}$ is the unpolarized radiance response of the OBM and $\mathbf{M}(\alpha)$ is the scan mirror Mueller matrix which depends on the incidence angle(s) $\alpha$. Depending on the scan mirror configuration in use (one or more mirrors with reflection planes at an angle), it can be the product of several Mueller matrices appropriately transformed to match their respective coordinate systems. The Mueller matrix of any of the scan mirrors can be calculated from the Fresnel equations, possibly taking into account multiple layers of material and knowing their complex refractive indices and thicknesses. The separation of the instrument into these two modules enables two separate calibration steps: On-ground, the calibration data which were obtained with particular scan mirror configurations can be modeled with uncontaminated mirrors to yield the OBM Mueller vector (Krijger, 2017). In-flight, the calibration measurements with a stable, unpolarized source (the sun or the internal White Light Source) can be used to determine the thickness and refractive index of a contaminant layer on each of the involved scan mirrors, assuming a fixed OBM. Realistically, the limited information contained in the combination of calibration measurements used only allows for the determination of a single, constant in time though wavelength dependent refractive index and the time dependent thicknesses on each of the scan mirrors. Additionally, an extra factor, called *M-Factor*, is introduced to indicate the degradation of the unpolarized OBM response. However, the separation of mirror and OBM degradation given the information contained in the calibration measurements is not entirely unambiguous, in the sense that to some degree both can describe the unpolarized calibration measurements equally well.

## 2.3 Nadir and limb scans

The relevant measurements discussed here are nadir and limb reflectance and polarization measurements. SCIAMACHY performs both observation modes in an alternating sequence. Envisat's sun-synchronous orbit crosses the descending node at 10 a.m., such that the sun is always to the left of the flight direction, with a minimum solar zenith angle (SZA) of about $20°$.

A nadir scan is carried out by pointing the ESM (Elevation Scan Module) mirror toward the surface and then continuously scanning from left to right, thereby covering elevation angles up to $32°$ and a total swath width of 960 km. During the scan, the detector signal is read out several times. The integration time depends on wavelength and SZA. The data considered here are all selected to have integration times of 0.25 s or less, such that 16 readouts per scan are available. The ground pixel size thus amounts to 60 km across track as determined by the integration time and 30 km along track determined by the length of the instrument slit of $1.8°$ and the motion of the spacecraft. The scan takes 4 s and is followed by a fast backward scan of 1 s. This sequence is typically repeated 13 times before switching to a limb scan.

During a limb scan, the ASM (Azimuth Scan Module) mirror is pointed approximately into the flight direction, toward the limb of Earth's atmosphere. The projection of the instrument slit is parallel to the horizon, providing an instantaneous field of





view of about 103 km horizontally and 2.6 km vertically. The light is reflected off the ASM mirror toward the ESM mirror which is set at an elevation angle of about $63°$. The reflection planes of both mirrors are at an angle to each other. In the scan mirror model, this angle is properly accounted for by rotating the respective coordinate systems. The ASM mirror performs a horizontal scan of about $\pm 10°$ around the flight direction at a fixed ESM elevation angle, starting approximately 3km below

the horizon. The horizontal scan takes 1.5 s, again covering a swath of 960 km. After the scan, the ESM elevation angle is increased by about 3.3 km and the horizontal scan proceeds in the other direction. Between 29 and 34 such horizontal scans are performed in one sequence, thus covering altitudes between about -3 and 93 km or higher. After the last scan, another ESM elevation step to an altitude of about 250 km is executed, and a dark measurement is performed for the last 1.5 s of the sequence. This dark measurement is typically used to correct the detector signals for potentially orbit phase dependent

dark current conditions. During a horizontal scan, the typical integration time is 0.375 s, resulting in 4 radiance profiles per limb scan. With Envisat flying at an altitude of about 790 km, the tangent points of a limb scan are about 3300 km ahead of the sub-satellite point. The measurement sequence is designed such that about 7 min after a limb scan, a nadir sequence is performed whose ground pixels overlap with the limb tangent points, thus enabling stereoscopic measurements. In Fig. 1, the scan mirror configurations and viewing geometries for the nadir and limb observation modes are depicted on the left and right, respectively.

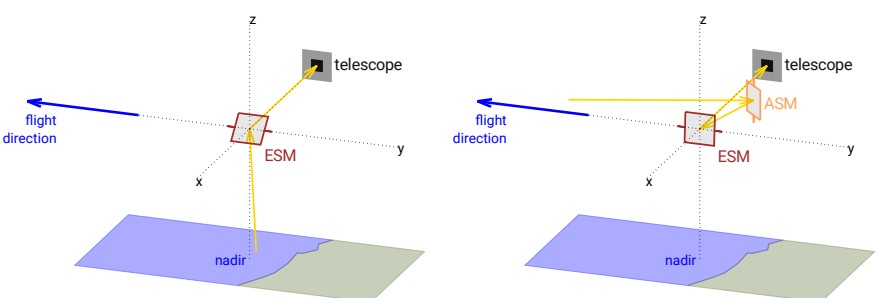

**Figure 1.** Sketch of the scan mirror configuration and viewing geometry during nadir (left) and limb (right) observations. In nadir mode, the ESM mirror is rotated to achieve a scan across track. In limb mode, the ASM mirror is rotated to achieve a scan across the flight direction, while the ESM mirror performs small steps to select a specific tangent height. Their reflection planes are rotated w.r.t. each other.

The true light path along the line-of-sight of a limb measurement depends critically on wavelength and the scene observed. Topography and horizontally inhomogeneous cloud or aerosol layers make observations in the troposphere highly variable. However, in the stratosphere, under relatively stable aerosol conditions, reflectance measurements are quite suitable for systematic studies. A possible issue persists in the contamination with spatial stray light, either from the sun or from bright

scenes below the tangent altitude. The measurements considered here are always done with the relative azimuth angle with the sun larger than $20°$, thereby minimizing the possible contamination with solar stray light. Earth shine stray light, however, is





clearly visible at tangent heights above 40 km and becomes dominant at altitudes larger than about 60 km. Below 40 km, its contribution is expected to be not larger than a few per cent.

Reflectances are obtained by normalizing limb or nadir radiances with the solar irradiance, which is measured once per day using a combination of a diffuser on the back side of the ESM and the ASM mirror.

## 2.4 SCIAMACHY Polarization Measurements

Generically, polarization can be measured by modifying the Mueller matrix in Eq. 4 such that the measured signal becomes sensitive to different linear combinations of the Stokes vector components. In order to derive all components unambiguously, at least 4 independent measurements per wavelength have to be carried out. The circular polarization component is typically negligible in atmospheric polarization, reducing the minimum number of required measurements to 3. These can be either carried out sequentially, or, as in the case of SCIAMACHY, simultaneously by employing a beam splitter. The beam split device in SCIAMACHY is the so-called *predisperser prism* which is located right after the telescope. It serves two purposes: The first is to spatially separate different spectral ranges of the incoming light in order to facilitate their diversion into SCIAMACHY's eight spectral channels. The second purpose is to split off a small fraction of the beam through internal reflection at the Brewster angle and thus generate an almost fully polarized beam which is directed towards the Polarization Measurement Devices (PMDs), a set of photodiode detectors sampling at a rate of 40 Hz and integrating over relatively wide spectral regions (100 nm to 200 nm) that roughly correspond to the central part of each of the main science channels. Figure 2 gives a schematic overview of the measurement principle. After passing the scanner module depicted on the left, the light enters the telescope with the instrument slit. The incidence on the first surface of the predisperser prism is nominal, thus no spectral diversion occurs inside it. Most of the light exits on the opposite side, which is tilted, and is refracted toward the science channels. The remaining part of the beam undergoes internal reflection under the Brewster angle, after which it is almost fully polarized with its polarization direction pointing into the paper (indicated by the crossed circle), or parallel to the instrument slit. After another internal reflection it exits the prism and is refracted towards the PMDs.

Table 1 lists the PMDs used for this investigation and their relevant properties. The average wavelength given there is approximate, it depends in detail on the shape of the measured spectrum. Also given in this table are the sensitivity to unpolarized light relative to that of the science channel ($\langle M_1 \rangle^{PD}$) and the three components of the OBM Mueller vector, all spectrally averaged. PMD 7 is special in the sense that its signal is not split off at the predisperser prism, but rather at the location of channel 5, and the $45°$-polarization is obtained by employing a polarization filter. Its spectral range is nearly identical with that of PMD 4. While PMD 7 has its largest sensitivity to light polarized at $45°$, i.e., $\mu_3 \approx 1$, the other PMDs are mostly sensitive to light polarized parallel to the direction of the instrument slit, hence $\mu_2 \approx -1$. Take note of the relatively large sensitivity to circular polarization, which in fact is unexpected given the design of the PMDs and poses a serious problem to the SCIAMACHY polarization measurements. Even though the atmospheric radiance is not circularly polarized, the scan mirrors will convert $45°$ linearly polarized light to circularly polarized light. The end-to-end Mueller vector of the PMDs thus contains a large, unwanted sensitivity to $45°$-polarization. A similar effect was discovered for the science channels as well. Since it was originally deemed sufficient to determine only $q$ and to correct the science channel radiance for this Stokes component only,





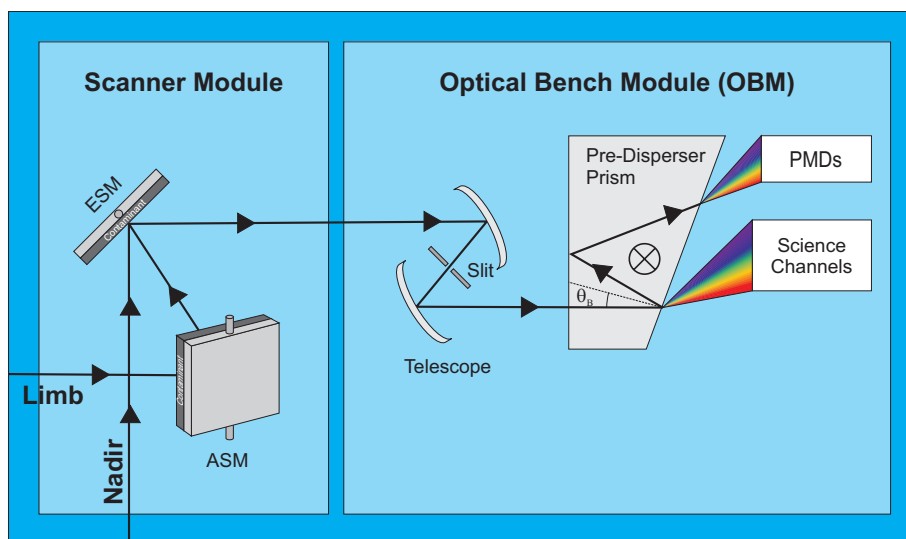

**Figure 2.** Sketch of the measurement principle of SCIAMACHY. The left part indicates the scanner module with different mirror configurations, their optical properties described by the scan mirror model. The scan mirrors can contain a contaminant layer. On the right, the optical bench module is depicted, which includes the telescope, the predisperser prism and the science channel and PMD detectors. The crossed circle indicates the polarization direction of the PMD beam after internal reflection inside the prism (into the paper). The optical train between the predisperser prism and the PMD and science channel detectors has been omitted, for details, we refer to Gottwald and Bovensmann (2011)

a measurement of the $u$-component was not foreseen. This is only possible for the wavelength region around 850 nm, where both PMD 4 and 7 are available. For the other regions, assumptions on the relationship between $q$ and $u$ or on the value of $u$ itself have to be made in order to be able to derive $q$ and to correct the measured signal adequately.

5   The polarization is determined from the ratio of the synchronized and integrated (over the exposure time of the science channel) PMD signal to the integrated (over the PMD spectral band) science channel signal:

$$S^{\mathrm{P}} = \mu_1^{PD} \cdot \sum_i S_i^{\mathrm{D}} M_{1,i}^{\mathrm{PD}} \frac{1 + \mu_{2,i}^{\mathrm{P}} q + \mu_{3,i}^{\mathrm{P}} u}{1 + \mu_{2,i}^{\mathrm{D}} q + \mu_{3,i}^{\mathrm{D}} u}, \quad \text{with} \tag{6}$$

$$M_{1,i}^{\mathrm{PD}} = \frac{M_{11,i}^{\mathrm{P}}}{M_{11,i}^{\mathrm{D}}} . \tag{7}$$

The sum goes over all pixels in the relevant spectral range, the superscripts $P$ and $D$ indicate PMD and science detectors,

10  respectively. The $\mu_i^{P,D}$ are end-to-end Mueller vector elements and vary with observation mode and scan angle. The factor $\mu_1^{PD}$ is an additional in-flight calibration factor that accounts for calibration offsets in the relative PMD to science channel response to unpolarized light. Assuming that the polarization and the polarization sensitivity varies sufficiently slowly with





wavelength, this equation can be further simplified:

$$P \equiv \mu_1^{PD} \frac{S^P}{S^D} \approx \frac{1 + \langle \mu_2^{\mathrm{P}} \rangle q + \langle \mu_3^{\mathrm{P}} \rangle u}{1 + \langle \mu_2^{\mathrm{D}} \rangle q + \langle \mu_3^{\mathrm{D}} \rangle u} \tag{8}$$

with

$$S^D = \sum_i S_i^{\mathrm{D}} M_{1,i}^{\mathrm{PD}} . \tag{9}$$

and

$$\langle \mu_n^{\mathrm{P,D}} \rangle = \frac{1}{S^D} \sum_i S_i^{\mathrm{D}} M_{1,i}^{\mathrm{PD}} \mu_{ni}^{\mathrm{P,D}} , \quad n = 2,3 \tag{10}$$

The intensity weighted spectrally averaged polarization sensitivities. The term $S^D$ is also called *virtual sum* and describes the expected PMD signal for zero polarization, given the science channel signal and the relative detector responses. The measured ratio $P$ of PMD signal to virtual sum is the *polarization signal* and should only depend on the polarization.

Given known calibration constants and a relationship between $q$ and $u$, or a known value for $u$, Eqs. 6 or 8 can be solved and result in a value for $q$ which can be interpreted both as the signal weighted average over the spectral width of the PMD and as its value at a representative wavelength[1]. This wavelength can be determined via

$$\langle \lambda \rangle = \frac{1}{S^D} \sum_i \lambda_i S_i \left( M_{1,i}^{PD} \right)_i ; . \tag{11}$$

On the other hand, if the polarization values at the representative wavelength are known, Eq. 8 enables the determination of the average $\langle \mu_n^{\mathrm{P}} \rangle$ for the PMDs and the unpolarized offset $C_{IB}$. In the next section, the details of the calibration approach are laid out.

## 3   Calibration Model and Data

The calibration approach taken here exploits a phenomenological relationship between the measured reflectance and the depolarization. In general it should be possible to find a maximum polarization value for any unique observation geometry defined by the line-of-sight elevation angle ($\theta_{LOS}$), the SZA ($\theta_0$) and the relative azimuth between sun and line-of-sight ($\phi$). The maximum value is typically given by pure Rayleigh scattering. Scattering off aerosol- or cloud droplets or the surface decrease the polarization in most cases, notable exceptions being for instance rainbow or glory effects generated by water or ice clouds, respectively. In clear sky conditions over the ocean, sun glint can add substantial polarization. This effect is largest close to the principal plane, when SZA and elevation angle are approximately equal and the relative azimuth is small. It depends on wind speed and direction, which influence the distribution, direction and form of waves. Two possible strategies emerge from these considerations: One would be to estimate the maximum polarization value for a given observation geometry and to devise a way to extrapolate the measurements to this point. This strategy resembles a modification and extension of the approach taken

---

[1]This is not quite true for the case of PMD 1, where the average wavelength of the PMD signal is around 350 nm, but the polarization does not vary monotonically around this point. The wavelength matching the average polarization is closer to $\sim 370$ nm.





in Krijger et al. (2004) to validate polarization measurements with GOME. The other strategy would be to establish an adequate model that emulates the reflectance vs. polarization relationship in an averaged sense. These two strategies are explored in detail in the following.

## 3.1 Data quality criteria

Before the actual analysis, it is necessary to consider in detail the data selection criteria. Rainbow or Glory effects and strong specular reflection destroy the assumed depolarization-reflectance relationships. Therefore, in nadir, the observation geometries where such effects are likely to occur are excluded. In limb, the data considered are typically above cloud layers, and specular reflection cannot be directly observed, such that these effects are considered to be negligible. However, in high latitude winter it is possible that Polar Stratospheric Clouds (PSCs) occur which cause highly variable reflectance and polarization. A simple PSC

detection algorithm, similar to the one described in von Savigny et al. (2005) was implemented in order to filter out data points with PSCs. In both measurement modes, data with high SZA larger than $80°$ and with single scattering angles $|\cos(\theta)| \geq 0.8$ are rejected. Also, data where the subsatellite point is inside the Southern Atlantic Anomaly (SAA) region are excluded, because highly energetic particles are likely to affect the detectors or readout electronics severely and create "hot signals". In addition, quality criteria on the state of the instrument are imposed. For instance, data shortly after a decontamination period

are not considered for this analysis.

### 3.1.1 The limiting model

The Vector Radiative Transfer Model (VRTM) SCIATRAN (Rozanov et al., 2014) can be used to calculate, for each individual data point, the limiting values for $q$ and $u$, i.e., the maximum possible polarization. For nadir over land and for limb the limiting model is considered to be multiple Rayleigh scattering over a dark (albedo = 0) surface. Over ocean, in addition to Rayleigh

scattering an isotropic Cox-Munk parametrization (Cox and Munk, 1954) with a wind speed parameter set to $5\ \mathrm{ms^{-1}}$ is used to model the bidirectional reflectance distribution function (BRDF). The parameters set to define the water leaving reflectance and white cap contributions are the default parameters given in SCIATRAN V3.5 (Rozanov and Rozanov, 2016). A correction for atmospheric refraction is employed for limb and nadir simulations. Nadir data were simulated with a pseudo-spherical geometry, i.e., the solar beam is modeled in a spherical shell atmosphere, the scattered beam in a plane parallel one. In limb,

a spherical atmosphere is assumed for both cases, although for technical reasons the multiply scattered part of the beam is approximated by a plane parallel atmosphere. This leads to approximation errors in reflectance and polarization, because the contribution of multiple scattering is overestimated.

Absorption by trace gases is modeled using a climatology for trace gas concentrations according to McLinden et al. (2010) for continuum absorbers and Sinnhuber et al. (2009) for line absorbers. Settings for the temperature dependence of $O_3$ cross

sections etc. are the default ones employed in SCIATRAN. Note that the exact modeling of atmospheric trace gas profiles is not too critical here, because in most cases the average PMD wavelengths are outside, or only within weak ($O_3$ or $H_2O$) absorption bands. The Stokes vector was calculated for fixed wavelengths representing the average PMD wavelengths, integrating over a typical band width of 1 to 5 nm. The RTM values have been generated in latitude bands of 10 degrees and for each month





of the year and are mapped into a regular grid in $(\theta_{LOS}, \theta_0, \phi)$. Eventually, by means of linear interpolation (in observation geometry only) a reference Stokes vector for the limiting RTM is available for each data point.

### 3.1.2 The extrapolation method

The data points for a given time period are now grouped according to the relevant scan angle (the position of the ESM mirror in nadir, the position of the ASM mirror for a fixed tangent height in limb) and in cells of $(q_{RTM}, u_{RTM})$ with a typical width of 0.01 in each direction (0.02 for limb), as a function of the RTM normalized average reflectance $R/R_{RTM}$ at the PMD central wavelength. The subscript $RTM$ stands for the appropriate limiting model. The average reflectance is defined here as

$$R = \pi \frac{1}{N} \sum_i \frac{I}{I_0} \qquad |\lambda_i - \lambda_C| \le \delta\lambda, \tag{12}$$

where $I$ is the measured intensity and $I_0$ the solar irradiance at each spectral pixel $i$ within a window of width $\delta\lambda$ around the central wavelength $\lambda_C$ given in table 1. The width $\Delta\lambda$ is between 1 nm and 5 nm, depending on wavelength, and $N$ is the number of valid spectral points inside this window. The $(q_{RTM}, u_{RTM})$-cells are small enough that the viewing angles in each vary only slightly, thus not mixing too different processes with different viewing geometries into one cell. In each cell, the variation of the polarization signal $P$ with $\delta R/R_{RTM} = R/R_{RTM} - 1$ can be fitted with a function of the form:

$$f\left(\frac{\delta R}{R_{RTM}}\right) = p_0 + \frac{p_1}{1 + p_2 \frac{\delta R}{R_{RTM}}} \tag{13}$$

with three fit parameters $p_i$. This functional form is phenomenological, but motivated by the expected behavior of depolarization induced by a *single* physical process, for instance the increase of surface albedo or of the aerosol concentration of a fixed aerosol type. The depolarization of each individual process would then be governed by the parameter $p_2$. In reality, multiple processes would contribute and cause considerable scatter. The underlying assumption here is that this scatter should decrease as the measured reflectance approaches the RTM limit since each possible process can only have a small effect, such that the data should converge to a well defined slope. The polarization signal corresponding to the maximum (RTM) polarization in the center of the cell is then

$$P(0) = f(R = R_{RTM}) \equiv P(q_{RTM}, u_{RTM}) = p_0 + p_1. \tag{14}$$

For this fit, the reflectance is corrected for polarization with the in-flight MMEs obtained from the scan mirror model and the on-ground OBM Mueller vector for the corresponding wavelengths. As in this step of the analysis the actual polarization values for each data point data are not available, the polarization values used in the correction are the RTM values themselves, i.e., each data point is corrected for the *maximum* polarization:

$$c_{RTM}^P = \frac{1}{1 + \mu_2 q_{RTM} + \mu_3 u_{RTM}}. \tag{15}$$

This means that the reflectance curve will be somewhat distorted towards the high reflectance tails. It has been verified, though, that this has no influence on the estimate of the limit towards the low reflectance values, where the correction should be closer to the truth.





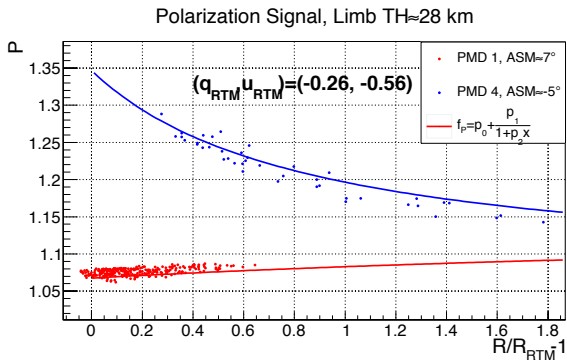

**Figure 3.** Examples for the *Extrapolation Method* for Limb, for a fixed cell of $(q_{RTM}, u_{RTM})$. Blue points are for PMD 4, red points for PMD 1, the corresponding curves show the fitted functions.

In figure 3 the method is demonstrated for the case of PMD 1 and PMD 4 in limb. Note that for the examples in the figure the viewing geometries for PMD 1 and 4 are not the same, despite the RTM polarization values being the same. Also, in order to get a sufficient amount and leverage of data for the fit, the data have to be collected over the period of at least a month. To avoid large extrapolation errors it is also vital to disregard cells where the minimum reflectance is too far away from the RTM limit, which happens very often at higher wavelengths in nadir over land due to the contribution of surface reflectance.

The fitted maximum polarization values fill a certain range in the $(q, u)$-plane which is determined by scan angle, scattering geometries and the period of time integrated over. Here, this period is normally one calendar year, except for 2002, when it starts only in August after the end of the commissioning phase and for 2012, when the mission ended in April. A typical example for the maximum polarization signal obtained in nadir, 2004, is given in figure 4, for PMDs 1 and 4.





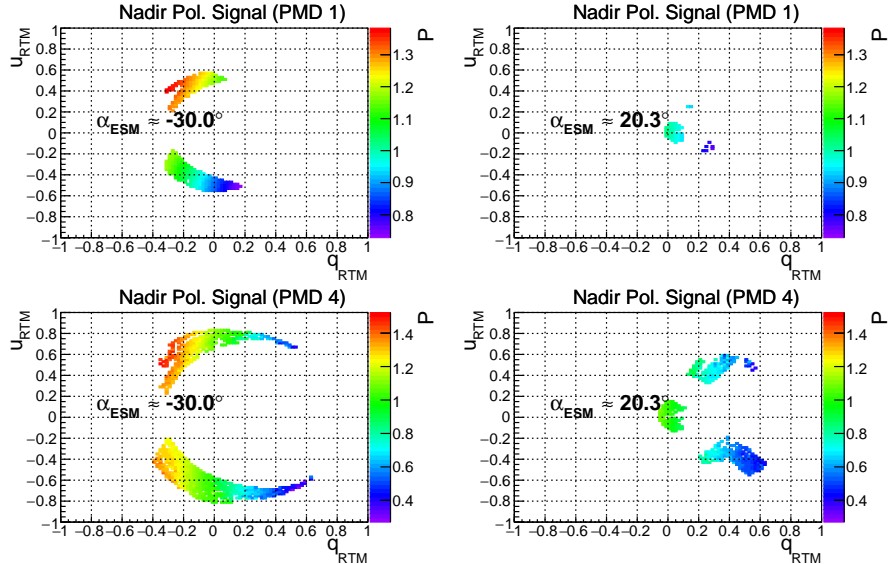

**Figure 4.** Polarization signals for nadir (rainbow color code) vs. $q_{RTM}$ ($x$-axis) and $u_{RTM}$ ($y$-axis) over ocean in 2004 from the *Extrapolation Method* for PMD 1 (top), PMD 4 (bottom) and for two different ESM mirror positions during the nadir scan (left: East, right: West).

### 3.1.3 The look-up table method

The second approach is only realized for limb, where a large simulation data set is available for a subset of the limb observation geometries, comprising a range of aerosol profiles with different types, different cloud layers and types and values for the Lambertian albedo between 0 and 0.8. The details on this simulation data set are already described in (Liebing et al., 2013). In order to make these simulations applicable to all limb observation geometries, the simulated data are once again grouped according to observation geometry and averaged over a smaller subset of all scenarios, including all but the highest values of stratospheric AOD which are unrealistic for the time period considered here. An arithmetic average of $q/q_{RTM}$ ($u/u_{RTM}$) of all scenarios is determined as a function of $R/R_{RTM}$, where the subscript $RTM$ refers to the limiting RTM value for pure Rayleigh scattering. After smoothing, this average is mapped as a function of $q_{RTM}$ ($u_{RTM}$), such that for each PMD, a two-dimensional look-up table (LUT) is available that delivers the depolarization in each Stokes component as a function of its maximum RTM value. The approach is illustrated in Fig. 5 for PMDs 1 and 4. On the left side, the individual simulation points and their averaged values for a fixed TH (28 km) and fixed viewing geometry are shown, the right side shows the two-dimensional LUT built from a range of viewing geometries. From this LUT, a value for $(q_{LUT}, u_{LUT})$ can be determined for each data point given in terms of $(q_{RTM}, u_{RTM}, R/R_{RTM})$:

$$p_{LUT} \quad = \quad p_{RTM} \left\langle \frac{p}{p_{RTM}} \right\rangle (p_{RTM}, R/R_{RTM}) \qquad p = q, u; \tag{16}$$





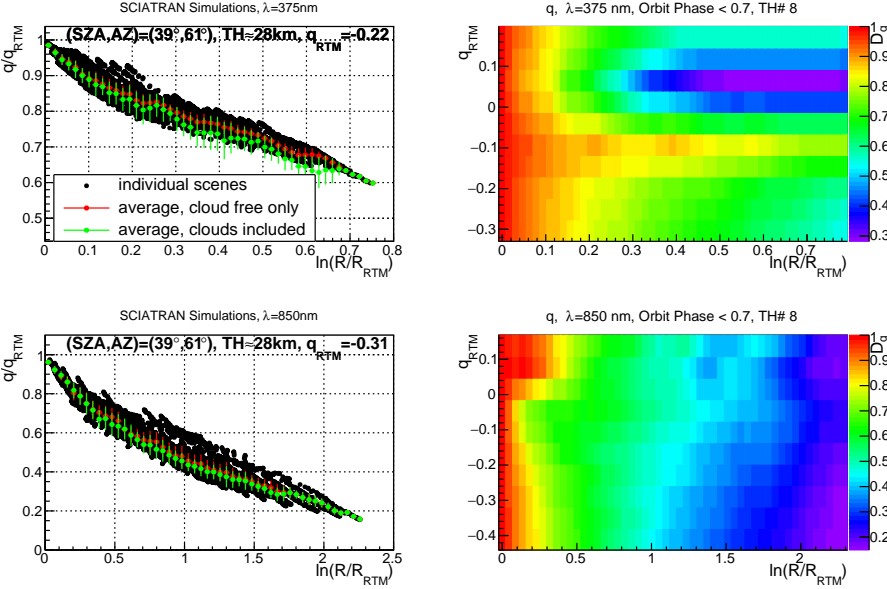

**Figure 5.** Example for the derivation of LUTs for the depolarization in limb geometry, $TH \approx 28$ km. Top is for 375 nm (i.e., PMD 1), bottom for 850 nm, i.e, PMDs 4 and 7. On the right the depolarization for a fixed geometry is plotted vs. $\log(R/R_{RTM})$, on the left the 2D LUTs are shown as $q/q_{RTM} = f(q_{RTM}, \log(R/R_{RTM}))$ in a rainbow color scale.

Also for this method, the reflectance values are corrected for polarization using Eq. 15, except that here the RTM polarization values are replaced by the LUT polarization values, and an iteration step is performed to arrive at the final value for $p_{LUT}$, given the polarization corrected reflectance.

These data are again collected in cells of $(q_{LUT}, u_{LUT})$ and in bins of ASM mirror angle and tangent height (TH) and aver-
aged over a year. Examples for the resulting distributions are given on the right side of Fig. 6, together with the corresponding results from the *Extrapolation Method* on the left, once again for PMDs 1 and 4. Comparing the *Extrapolation-* and *LUT-Method* values it is evident that the sampling is much denser for the *LUT-Method*, but that the range of $(q, u)$-values covered is larger with the *Extrapolation-Method* – naturally because the *Extrapolation-Method* by definition always selects the maximum possible polarization values. The *LUT-Method* is highly dependent on the assumptions used in the simulation of the different
scenarios, e.g., aerosol properties and profiles, and on the scenario selections and averaging method applied to prepare the LUTs. Resulting calibration values may therefore be considerably model dependent. The *Extrapolation-Method* on the other hand is only marginally model dependent since the basic mechanisms of Rayleigh and ocean surface scattering are relatively well known and can be accurately computed. The potential pitfall lies in the assumption on how representative the limiting polarization condition is and how well it can be reproduced by the fit with Eq. 13. For each individual case the resulting errors
can hardly be estimated since many different effects may cancel or enhance each other, but the comparison of the calibration parameters obtained by the two methods can provide a viable estimate of the typical uncertainties involved in this analysis.





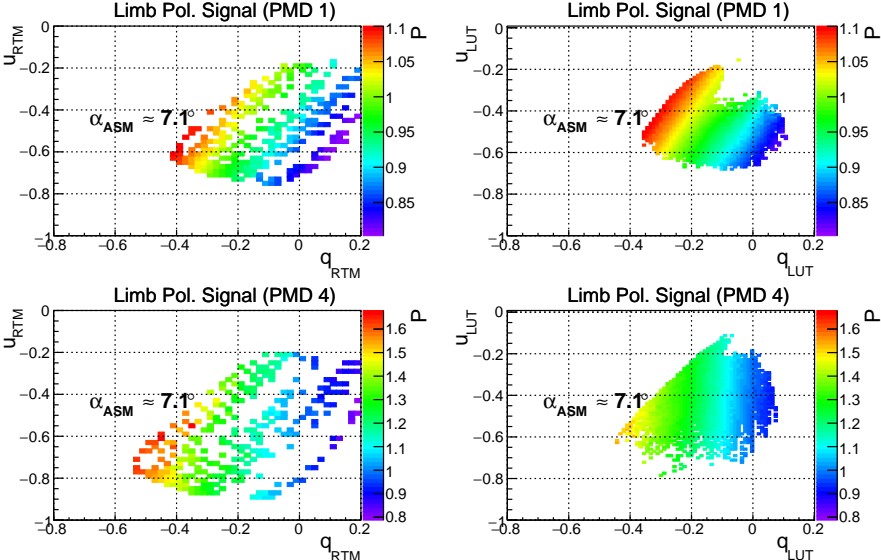

**Figure 6.** Polarization signals for limb (rainbow color code) in 2004 vs. $q_{RTM(LUT)}$ ($x$-axis) and $u_{RTM(LUT)}$ ($y$-axis) , $TH \approx 28km$ from the *Extrapolation Method* (left) and the *LUT Method* (right) for PMD 1 (top), PMD 4 (bottom), for an ASM mirror angle of about $+7°$.

## 3.2 The Polarization Feature at 350 nm

With the polarization signal as defined in Eq. 8, i.e., the ratio of PMD to science channel signals, it is only possible to derive the in-flight MMEs for the PMDs. There is no independent information on the science channel MMEs. A particularly strong polarization feature in SCIAMACHY's channel 2, located at around 350 nm, opens up the possibility of deriving information

5   about the in-flight polarization sensitivity at this wavelength, and such enables a direct comparison of science channel and PMD 1 behavior at almost the same wavelength. Figure 7 shows the polarized end-to-end MMEs (nadir and limb) for Channel 2 as a function of wavelength. The properties of this feature indicate the presence of a so-called Wood's Anomaly (see, e.g., Maystre (2012)) of the grating. The feature at 350 nm was recognized as a potential noncompliance with respect to the instrument specification. However, the high throughput of this grating resulted in its selection nevertheless. Its sensitivity to polarization

10   provides independent information on the polarization response in Channel 2, but may contribute to noise or bias in trace gas retrievals if not perfectly accounted for.

    The polarization sensitive signal can be constructed by comparing two relatively close spectral points with very different polarization sensitivity. An obvious choice would be a point around 350 nm close to the minimum of $\mu_2$, and another one outside the feature at 370 nm where both MMEs are relatively small. Points to the left of the feature are too sensitive to $O_3$





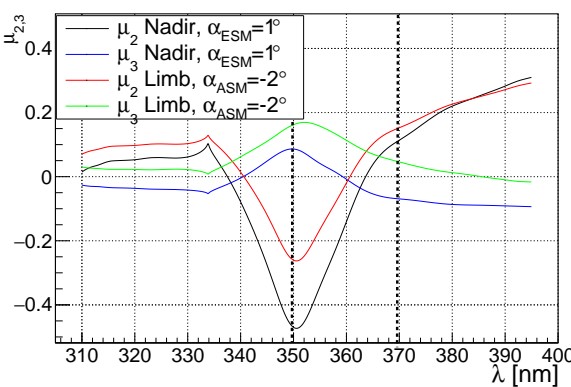

**Figure 7.** Polarized end-to-end MMEs ($\mu_{2,3}$) for nadir (black: $\mu_2$, blue: $\mu_3$) and limb (red:$\mu_2$, green: $\mu_3$) calculated with the scan mirror model for August 2002. The MMEs are given in the atmospheric Stokes frame. The thick dashed vertical lines indicate the two spectral points used to build the polarization signal (see text).

concentrations and therefore not considered. The polarization signal is:

$$P_{350}\left(\frac{R(370\,\mathrm{nm})}{R_{RTM}(370\,\mathrm{nm})}\right) \equiv \frac{R(350\,\mathrm{nm})/R(370\,\mathrm{nm})}{R_{RTM}(350\,\mathrm{nm})/R_{RTM}(370\,\mathrm{nm})} \tag{17}$$

$$= C\left(\frac{R(370\,\mathrm{nm})}{R_{RTM}(370\,\mathrm{nm})}\right)\frac{1+\mu_2^{350}q(350\,\mathrm{nm})+\mu_3^{350}u(350\,\mathrm{nm})}{1+\mu_2^{370}q(370\,\mathrm{nm})+\mu_3^{370}u(370\,\mathrm{nm})}. \tag{18}$$

The factor $C$ accounts for a possible calibration offset as well as a change of the spectral slope between 350 nm and 370

5 nm with atmospheric or surface conditions. This is in contrast to the PMD to science channel polarization signal, which by definition depends only on the polarization. For $P_{350}$, this is only the case for the point where $R(370\mathrm{nm}) = R_{RTM}(370\mathrm{nm})$, i.e., at the RTM limit. This also implies that the *LUT-Method* cannot be applied for this polarization signal.

### 3.3 Derivation of in-flight Mueller matrix elements and their errors

#### 3.3.1 Fit of end-to-end MMEs

10 The result of the previous analysis step are three dimensional distributions of the polarization signal in $(q, u, \alpha)$, which can be modeled as (see also Eq. 8)

$$P(u, q, \alpha) = \langle\mu_1\rangle(\alpha)\frac{1+\langle\mu_2^P\rangle(\alpha)q+\langle\mu_3^P\rangle(\alpha)u}{1+\langle\mu_2^D\rangle(\alpha)q+\langle\mu_3^D\rangle(\alpha)u} \tag{19}$$

where $\alpha$ is the scan angle and

$$\langle\mu_i\rangle(\alpha) = p_{i0} + p_{i1}\alpha + p_{i2}\alpha^2, \qquad p_{i2} \equiv 0 \text{ for limb.} \tag{20}$$

15 Due to the limited range of $(q, u)$ values covered it is not possible to simultaneously fit all five MMEs. The aim is instead to fit the PMD parameters and the unpolarized scale factor $\langle\mu_1\rangle$ in a linear fit. This is achieved by correcting the polarization signal





for the science channel contribution by simply multiplying each cell value by a factor

$$C^D = 1 + \langle \mu_2^D \rangle (\alpha) q_{RTM,LUT} + \langle \mu_3^D \rangle (\alpha) u_{RTM,LUT}. \tag{21}$$

For the Channel 2 feature $\mu_i^{350}$ corresponds to $\mu_i^P$, and the $\mu_i^D$ are to be replaced by the MMEs at 370 nm, $\mu_i^{370}$. The RTM polarization values at 370 nm have to be used for the correction of the reflectance.

The fit is performed as a weighted least squares fit. The fit parameters are the $p_{ij}$ in Eq. 20. The statistical errors of the fit parameters and their correlations are determined by the extrapolation fit error (*Extrapolation-Method*) or by the error of the mean (*LUT-Method*) in each cell and the density and leverage of the 2D $(q, u)$ distribution. Figure 4 can already give an impression about the uneven distribution of data cells for different scan angles in nadir, and in Fig. 6 it should be pointed out that the limb data cover mostly only one quadrant of the possible $(q, u)$-range. Statistical uncertainties and correlations

are therefore expected to increase with scan angle in nadir, and to be larger in limb than in nadir. However, the statistical uncertainties are negligible compared to the systematic ones. This is indicated already by the typical reduced $\chi^2$ of the fit, which is significantly above 2 for all but PMD 1 and the Channel 2 feature.

### 3.3.2 Systematic Errors

In order to obtain a realistic estimate of the systematic uncertainties related to this analysis it is possible to vary some of its

critical assumptions and compare the results. The normalization uncertainty in the calibration of the reflectance would alter the $x$-axis of the polarization-reflectance curve and shift the RTM limit point, in all observation modes and in both methods. There is no official estimate yet available for the calibration uncertainty with the scan mirror model. Based on comparisons with previous Level-1 data versions and of nadir data with MERIS data, on the value fitted relative PMD to science channel calibration factor $\mu_1$ and considering possible stray light contamination in limb, a value of $\delta R/R \approx \pm 5\%$ appears to cover the

expected order of magnitude quite well. The systematic uncertainty due to the calibration error is estimated by shifting the reflectance of each data point by $\delta R$ and then repeating the entire chain of fits.

     In nadir over ocean, an obvious error source is the assumption on the wind speed for the ocean BRDF model. This wind speed is not meant to be representative, but rather the lower boundary against which the data are measured. The choice of $5 \, \mathrm{m \, s^{-1}}$ was optimized by comparing RTM normalized reflectance distributions for different wind speeds. However, slightly

lower or higher wind speeds seemed also viable. A wind speed related uncertainty for nadir can therefore be estimated from replacing the RTM reference values with those for wind speeds of $3 \, \mathrm{m \, s^{-1}}$ and $7 \, \mathrm{m \, s^{-1}}$, then again repeating the entire analysis chain and comparing the results.

     For limb data it is not directly possible to vary the model assumptions because of the prohibitively large effort to generate a comprehensive RTM data base. It is conceivable, though, that by investigating the results obtained for a range of tangent

heights could deliver a proxy for the model uncertainty. This is because certain model parameters maybe more appropriate at one TH than at another, or because contributions of albedo, stray light etc. change with TH. The limb data are therefore analyzed for a range of THs between 25 km and 35 km and compared to the reference value at 28 km.





For each of these systematic error classes (reflectance normalization error, BRDF model error in nadir and TH variation in limb) a systematic uncertainty with respect to the reference model and data is derived by computing the standard deviation and the covariance as a function of scan angle for the end-to-end MMEs. The covariance between the derived $\mu_2$ and $\mu_3$ is also computed. The total covariance matrix is obtained as the squared sum of the statistical and each considered class of systematic

error matrices. The combined covariance matrix is dominated by the systematic contribution, with strong (anti-)correlation prevailing between different scan angles for one MME, or between $\mu_2$ and $\mu_3$ at a given scan angle. Those correlations are close to $\pm 100\%$ in many cases and the correlation matrix turns out to be numerically degenerate when used in subsequent fits of the in-flight MMEs with the retarder model (see below). The covariance matrix is therefore "regularized" by scaling the off-diagonal elements by a factor 0.8, thus allowing additional, unknown, error components to destroy the almost perfect

correlation between fit parameters.

### 3.3.3 Nadir results

The fit results for the nadir end-to-end MMEs for the PMDs 1, 2 and the Channel 2 feature are displayed in Fig. 8. The curves

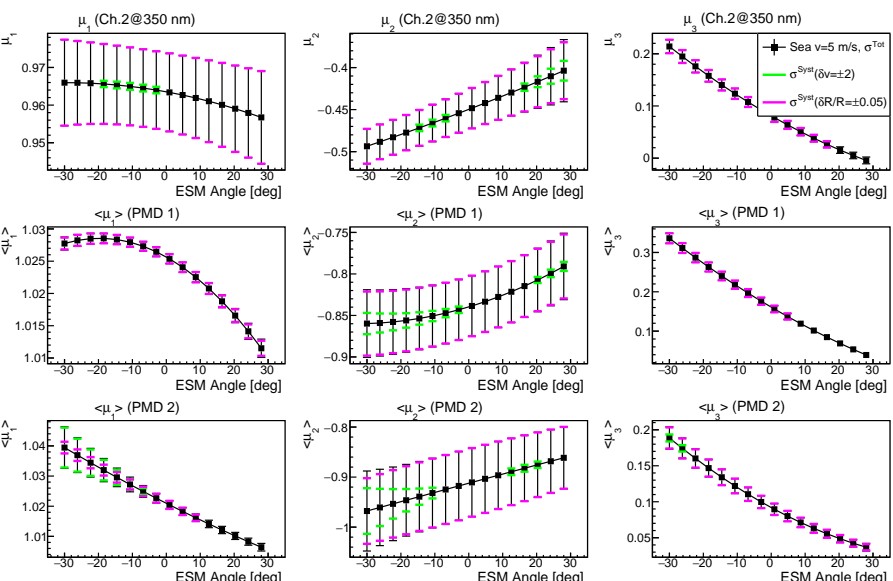

**Figure 8.** Final results for end-to-end MMEs vs. ESM angle for Nadir over ocean in 2004, from the *Extrapolation Method*. Top: Channel 2 feature, middle: PMD 1, bottom: PMD 2, left: $\langle \mu_1 \rangle$, center: $\langle \mu_2^P \rangle$, right: $\langle \mu_3^P \rangle$. Black data points are the results with total errors, magenta and green error bars indicate the contribution from each class of systematic errors.

with the error bars depict the final results to be used for further analysis, these are the data over ocean. The different color error bars indicate the individual contribution from the reflectance normalization (magenta) and wind speed (green) variations to the

15 total error. In the UV, the dominating source of error is the calibration uncertainty, while the wind speed uncertainty takes over at higher wavelengths. The results shown here were fitted to the nadir data over ocean only. The results for the data over land





differ systematically from the data over ocean in most of the parameters, with increasing differences the higher the wavelength. In general, the data over land have to be regarded as unreliable due to their intrinsic sparsity – because of the above mentioned extrapolation errors as well as a sampling bias in $u$. The sampling is problematic because positive $u$-values prevail in Northern latitudes and negative values in the Southern hemisphere, were in general less land mass exists. This is why for this analysis

here, the data over land are disregarded.

### 3.3.4 Limb results

The fit results for the limb end-to-end MMEs for the PMDs 1, 2 and the Channel 2 feature are displayed in Fig. 9. Here, the

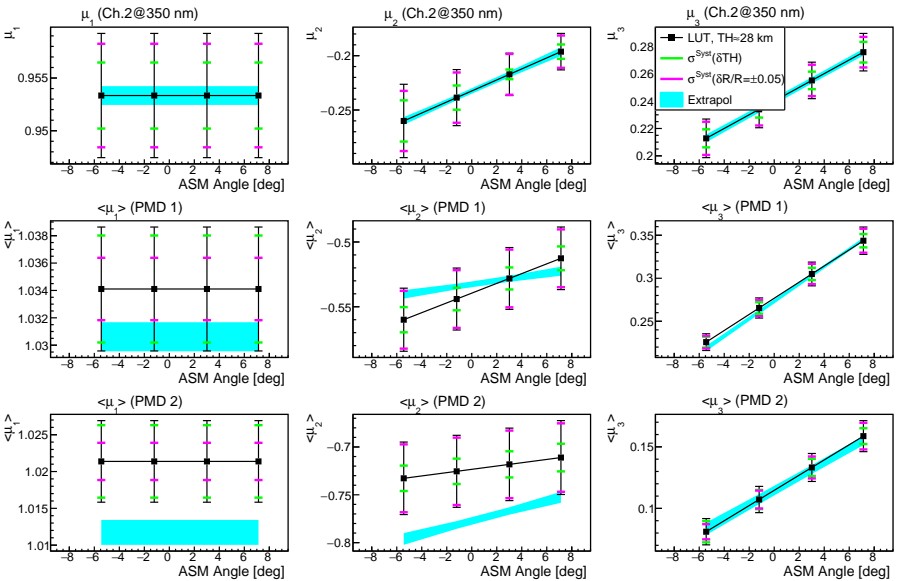

**Figure 9.** Final results vs. ASM angle for Limb at a TH of 28 km in 2004, from *LUT Method* (PMDs) and *Extrapolation Method* (Channel 2). Top: Channel 2 feature, middle: PMD 1, bottom: PMD 2, left: $\langle \mu_1 \rangle$, center: $\langle \mu_2^P \rangle$, right: $\langle \mu_3^P \rangle$. Black data points are the results with total errors, magenta and green error bars indicate the contribution from each class of systematic errors. The light blue band shows the result from the Extrapolation Method with statistical errors only.

green error bars indicate the contribution of TH variation, the magenta ones are again the contribution of the normalization uncertainty. For the UV-VIS regions considered here, both contributions are similar in magnitude. The light blue band is

the result from the *Extrapolation Method* (statistical errors only). For PMD 1 and Channel 2, the results are well within the systematic errors of the *LUT Method*, while for PMD 2 the difference is slightly larger, at least for $\mu_2$. In general the differences between the two methods increase with wavelength, as expected because intrinsic errors of both methods increase.

For the fits to the limb data, the unpolarized calibration factor $\mu_1$ is assumed to be independent of scan angle. The unpolarized calibration parameter constitutes an important in-flight correction for the polarization determination, but is treated as a nuisance

parameter in this analysis. It may be scan angle dependent at short wavelengths, where the degradation parameters of the mirror





model may introduce errors, but at higher wavelengths it is more likely to be constant. For the relatively small ASM scan angle range relevant for the limb data, leaving $\mu_1 = const.$ warrants better stability of the fit parameters, especially at the higher wavelengths. For the wavelengths relevant for Fig. 9, letting $\mu_1(\alpha) \neq const.$ has only a small effect within the systematic errors.

### 3.3.5 Comparison to the scan mirror model

For both observation modes it was observed that the results become increasingly unreliable with increasing wavelength. For the following comparisons and analysis we therefore concentrate only on PMDs 1 ($\sim 350$ nm), PMD 2 ($\sim 480$ nm) and the feature in Channel 2 at 350 nm. With the mirror model, the thickness and refractive index of the contaminant layers on the ESM and ASM mirrors can be determined from (solar) in-flight calibration data. Figure 10 shows the parameters of the current version of the degradation parameters, which is also to be used for future Level-1 reprocessing. The ESM contaminant thickness is

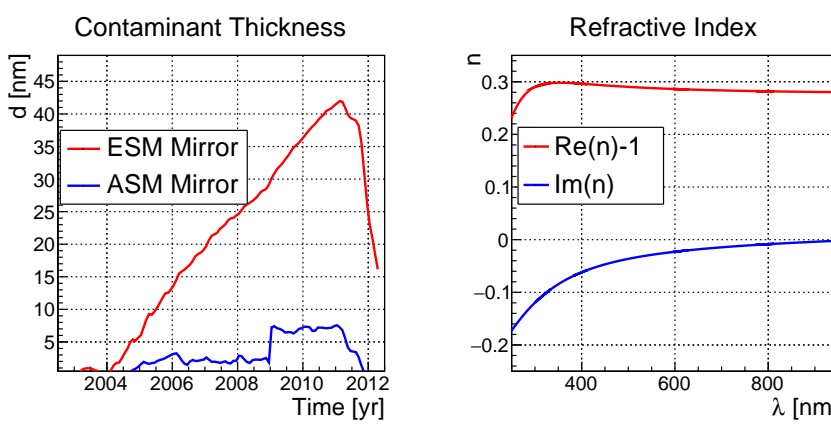

**Figure 10.** Left: Time dependence of contaminant layer thicknesses as determined from in-flight calibration data, red: ESM mirror, blue: ASM mirror. Right: The real (red, $Re(n) - 1$ is plotted) and imaginary (blue) part of the refractive index of the contaminant vs. wavelength.

increasing steadily until about 2011, and then experiences a steep drop. The ASM contaminant thickness is rather small and almost constant, increases abruptly after a decontamination procedure in January 2009 and also drops steeply after 2011. The refractive index is assumed to be constant in time and the same for all mirrors. Its imaginary part is largest in the UV. Due to this enhanced absorption of UV light by the contaminant layer, the effect of mirror degradation is expected to be largest in the UV region.

The scan mirror model can be used to compute the end-to-end MMEs for each observation mode, scan angle and point in time. The end-to-end Mueller vector according to Eq. 5 is the product of the OBM vector with the scan mirror Mueller matrix:

$$\boldsymbol{\mu}_{N,L}(\alpha) = \boldsymbol{\mu}^{OBM}\mathbf{M}(\alpha), \tag{22}$$

where the subscript $N$ and $L$ stand for nadir and limb, respectively. For the comparison, the mirror model values for the spectrally resolved PMD MMEs for the PMDs are averaged over wavelength using representative spectra for each month. As





the wavelength dependence of the PMD MMEs is very slow or even constant, the averaging does not have a large impact, though.

Figure 11 shows how the scan mirror model compares to the nadir MMEs and Fig. 12 to the limb MMEs for the duration of the mission. For nadir, the mirror model predicts rather constant values of $\mu_2$ over time and a significant increase in $\mu_3$. The

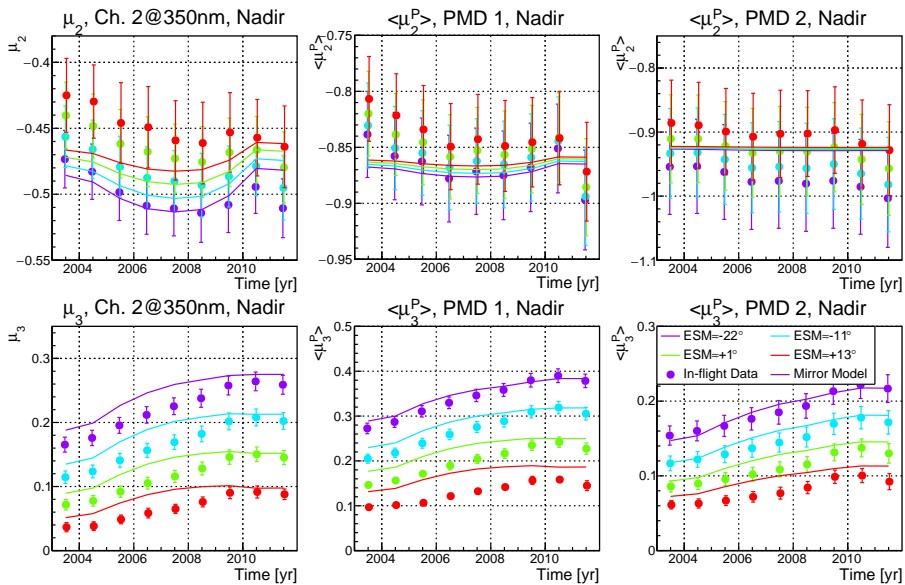

**Figure 11.** Time dependence of in-flight end-to-end Mueller matrix elements $\mu_2$ (top) and $\mu_3$ (bottom) for nadir for four selected ESM angles from measurements (data points) and the scan mirror model (lines). The error bars include statistical and total systematic errors. Left: Channel 2 feature, middle: PMD 1, right: PMD 2. The color code signifies the ESM scan angle bin (-22°, -11°, +1°, +13°), increasing from magenta over light blue and green to red.

values for $\mu_2$ reproduce the data quite well, although there seems to be a small but systematic decrease in the data during the first two thirds of the mission. Note that due to the systematic nature of the in-flight errors, the interannual variation may be highly correlated, such that trends may be significant despite the large individual errors. The larger scan angle dependence in the data on the other hand may not be significant. The data follow the predicted trend in $\mu_3$ approximately, but in the UV there is also a distinct offset, even close to the start of the mission. The differences decrease with wavelength.

In limb the situation is very different. The mirror model predicts a significantly smaller $\mu_3$ and the magnitude of $\mu_2$ seems to be overestimated. The time dependence of the data is not reproduced, as difference grow with time. Comparing to the right panel of Fig. 10 it is tempting to suspect a correlation of the observed trends in the differences between data and model with those of the contaminant thicknesses. A direct relationship is not straight forward to discern, though. The large offset at the start of the mission, in particular in limb, cannot be related to possible errors in the degradation parameters, as the contamination at

this point in time is practically nonexistent. The reason for this early discrepancy has to lie in the OBM.



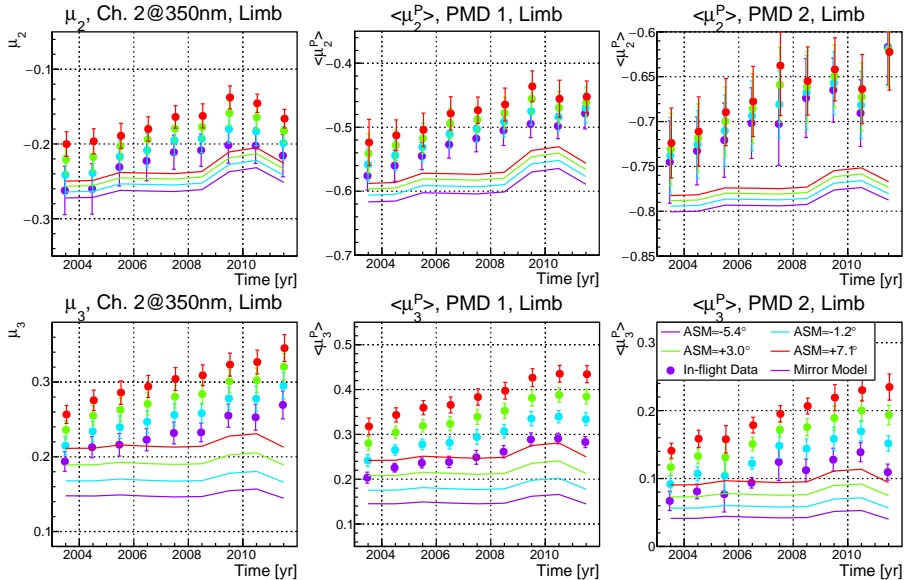

**Figure 12.** Time dependence of in-flight end-to-end Mueller matrix elements $\mu_2$ (top) and $\mu_3$ (bottom) for limb at $TH \approx 28$ km for four selected ASM angles from measurements (data points) and the scan mirror model (lines). The error bars include statistical and total systematic errors. Left: Channel 2 feature, middle: PMD 1, right: PMD 2. The color code signifies the ASM scan angle bin ($-5°$, $-1°$, $+3°$, $+7°$), increasing from magenta over light blue and green to red.

## 4 Interpretation of results

In this section, the results for the in-flight MMEs presented above will be interpreted with respect to a change of OBM polarization sensitivities between the on-ground calibration measurements and in-flight conditions. Already the on-ground calibration revealed an unexpectedly large sensitivity to $45°$-polarization end-to-end in nadir, mainly for the PMDs, but also

for the science channels. The on-ground calibration campaign was performed for fixed OBM- and detector temperatures close to the in-flight temperature settings. But some dedicated measurements were done with varying OBM temperatures, where polarization sensitivities were found to be temperature dependent. The peculiar wavelength- and temperature dependence thus pointed to temperature-induced stress birefringence as the likely physical cause (Snel, 1999). The predisperser prism, being the only optical element in the light path of both PMDs and science channels aside from the telescope, would serve as a

viable originator of this phenomenon. Even though the prism mount was designed to be stress free, it may be possible that through the mechanical stress generated by the presence of temperature gradients inside the instrument, birefringence can be generated. It is also possible that a small residual birefringence from the manufacturing process of up to $5\,\mathrm{nm\,cm^{-1}}$ (Keller, 2001) can be further enhanced due to stress. Under the microgravity conditions in-flight, stress may distribute differently, such that the experienced birefringence may change accordingly. Several analyses (Snel, 1999; Slijkhuis and Frerick, 1999)

of the on-ground data arrived at different results for the parameters associated with the alleged phase shift, based on different presumptions, definitions and data. Here, we present an analysis of the measured in-flight MMEs as well as the revised on-





ground OBM MMEs performed consistently within the frame work of the scan mirror model in order to test the hypothesis of an unintended stress induced phase shift inside the predisperser prism.

## 4.1 Retarder model

After the scanner module, the light beam enters the telescope where it undergoes reflections off the two parabolic aluminum
mirrors under a small angle and within the same plane. Under these conditions the polarization sensitivity of the telescope can be assumed to be negligible. A possible loss of reflectivity due to degradation would be common to both PMDs and science channels and not be visible in the measured polarization signal. After the telescope, the light enters the predisperser prism, which consists of fused silica glass. It travels for about $1.5\,\mathrm{cm}$ before a part of the beam is internally reflected at an angle close to the Brewster angle ($\sim 34°$ for fused silica) and emerges nearly 100% polarized to be directed towards the PMDs after
yet another internal reflection, while the major, still partially polarized, part of the beam is transmitted towards the science channels (see Fig. 2 above). Thus, assuming the telescope does not affect the polarized components of the OBM vector, the scan mirror model can be very simply extended by introducing the Mueller matrix for a linear retarder to represent the part of the predisperser prism before the Brewster reflection that can be influenced by birefringence:

$$S = I M_{11}^{OBM} \boldsymbol{\mu}^{OBM} \mathbf{R}(\delta, \theta) \mathbf{M}(\alpha)(1, q, u, v)^T. \tag{23}$$

where $\mathbf{R}$ is the Mueller matrix of a linear retarder

$$\mathbf{R}(\delta, \theta) = \begin{pmatrix} 1 & 0 & 0 & 0 \\ 0 & c_2^2 + s_2^2 \cos\delta & c_2 s_2 (1 - \cos\delta) & s_2 \sin\delta \\ 0 & c_2 s_2 (1 - \cos\delta) & s_2^2 + c_2^2 \cos\delta & -c_2 \sin\delta \\ 0 & -s_2 \sin\delta & c_2 \sin\delta & \cos\delta, \end{pmatrix} \tag{24}$$

and

$$
\begin{aligned}
c_2 &\equiv \cos 2\theta \text{ and} \\
s_2 &\equiv \sin 2\theta. 
\end{aligned}
\tag{25}
$$

Depending on the retarder angle $\theta$, the linear retarder simultaneously rotates the plane of linear polarization and converts linear into circularly polarized light (or vice versa), depending on the retardance $\delta$. At $\theta = 0°$ or $90°$, the cross talk between $U$ and $V$ is maximized, while at $\theta = \pm 45°$ the conversion takes place entirely between $Q$ and $V$. In terms of a birefringent crystal or slab of material, $\theta$ describes the direction of the optic axis w.r.t. which the incoming beam is split into an ordinary (polarization direction perpendicular to the optic axis) and extraordinary ray (polarization direction perpendicular to that of the ordinary ray)
and $\delta$ is the retardance of the slower ray relative to the faster (typically the ordinary) ray.

The retardance in a slab of thickness $d$ is

$$\delta = \frac{2\pi d}{\lambda}(n_e - n_o). \tag{26}$$





The actual *birefringence* is the difference between the indices of refraction for the extraordinary and ordinary ray, $n_e - n_o$. For stress induced birefringence, this can be related to

$$B \equiv n_e - n_o = R(\lambda)(\sigma_1 - \sigma_2) \equiv \frac{n(\lambda)^3}{2}(q_{11} - q_{12})(\sigma_1 - \sigma_2), \tag{27}$$

where $\sigma_{1,2}$ are the stresses along the polarization directions of the ordinary and extraordinary rays, and $q_{11}$ and $q_{12}$ are the components of the strain-optic tensor. The refractive index $n$ is wavelength dependent. $R$ is called stress-optic constant and also depends on wavelength (Sinha, 1978):

$$R(\lambda) = R(\lambda_0)\left[\frac{n(\lambda_0)}{n(\lambda)}\right]\left[\frac{\lambda^2}{\lambda_0^2}\right]\left[\frac{\lambda_0^2 - \lambda_1^2}{\lambda^2 - \lambda_1^2}\right]\left[\frac{\lambda^2 - \lambda_2^2}{\lambda_0^2 - \lambda_2^2}\right], \tag{28}$$

with $\lambda_1 = 121.5$ nm, $\lambda_2 = 6900$ nm, and $\lambda_0$ a normalization wavelength. For $\lambda_0 = 633$nm, $R = 35 \pm 1\,\mathrm{nm\,cm^{-1}\,MPa^{-1}}$ (Priestley, 2001). Therefore it is possible to build a model that in principle allows to determine the retarder matrix for all wavelengths from a measurement at a single wavelength. Here, $\lambda_0$ will be set to $300$ nm. The retardance at any other wavelength can then be modeled as:

$$\delta(\lambda) = \delta(\lambda_0)\frac{\lambda_0}{\lambda}\frac{R(\lambda)}{R(\lambda_0)}. \tag{29}$$

The retarder angle $\theta$ depends on the direction of the applied stress. The type of stress discussed here may be distributed non-uniformly across the prism such that different parts of the beam experience different rotations. Since the relevant phase shift occurs *before* the Brewster reflection (PMDs) or *before* the exit from the prism (science channels) and therefore also *before* wavelength dispersion, the averaged (over the beam cross section) retarder angle can be assumed to be wavelength independent.

## 4.2 Fit of retarder parameters

Based on Eqs. 23 and 22, the vector of retarder parameters $\mathbf{\Theta} = (\delta(\lambda_0), \theta)$ can be found in principle by minimizing the difference between a modeled and measured end-to-end Mueller vector:

$$\chi^2_{N,L} = \left(\boldsymbol{\mu}^{Meas}_{N,L} - \mathbf{X}(\mathbf{\Theta})\right)^T \cdot \boldsymbol{\Sigma}^{-1} \cdot \left(\boldsymbol{\mu}^{Meas}_{N,L} - \mathbf{X}(\mathbf{\Theta})\right); \tag{30}$$

with the instrument model

$$\mathbf{X}(\mathbf{\Theta}) \equiv \boldsymbol{\mu}_{L,N}(\alpha, \mathbf{\Theta}) = \boldsymbol{\mu}^{OBM}\mathbf{R}(\delta, \theta)\mathbf{M}(\alpha). \tag{31}$$

The vector of measurement points $\boldsymbol{\mu}^{Meas}_{N,L}$ contains the measured $\mu_2$ and $\mu_3$, each for four selected scan angles in nadir and limb, respectively. The covariance matrix $\boldsymbol{\Sigma}$ is determined from the total (statistical and systematic) error as described in section 3.3.2.

### 4.2.1 Confidence regions

Instead of directly combining the nadir and limb data, and data for different detectors, into one measurement vector, $\chi^2$ can be computed for each separately and the resulting distribution as a function of $(\delta(\lambda_0), \theta)$ can be used to evaluate the validity




of the retarder model in terms of its intrinsic consistency. In an ideal case, the best fitting retarder parameters would describe nadir and limb data, and each detector, equally well. In a statistical sense this means that the regions with

$$\Delta\chi^2 = \chi^2(\mathbf{\Theta}) - \chi^2_{Min} < \Delta\chi^2_{Max}(p). \tag{32}$$

for given confidence level $p$ should overlap for individual measurement modes and detectors. The confidence region $\mathbf{\Theta}_p$ in-

cludes all points $\mathbf{\Theta}$ with probability $\mathcal{P}(\chi^2 > \chi^2(\mathbf{\Theta}, \nu)) = 1 - CDF(\chi^2, \nu) > 1 - p$ for a $\chi^2$-distribution with $\nu$ degrees of freedom with the corresponding cumulative distribution function (CDF). Here $\nu$ corresponds to the number of considered fit parameters, in this case, $\nu = 2$ (Press et al., 2007). Strictly, this interpretation is only valid if the errors were purely statistical and Gaussian distributed. The systematics dominated errors of the in-flight data are neither, still, this approach seems the only practically feasible way to check consistency between independent measurements and combine them in order to maximize

information content and to estimate errors on the fit parameters and derived values. This approach is demonstrated in the following.

First, Fig. 13 shows the $\chi^2$-distribution for PMD 1, nadir (left panel) and limb (*LUT-Method*, right panel) in 2004. The parameter range has been restricted to $|\delta| \leq 45°$ and $\theta < 90°$ due to symmetries in the retarder matrix and because the $\chi^2$ for retardances outside the $\pm 45°$ limit increases even further. Inside the considered parameter space, $\chi^2$ varies by several orders

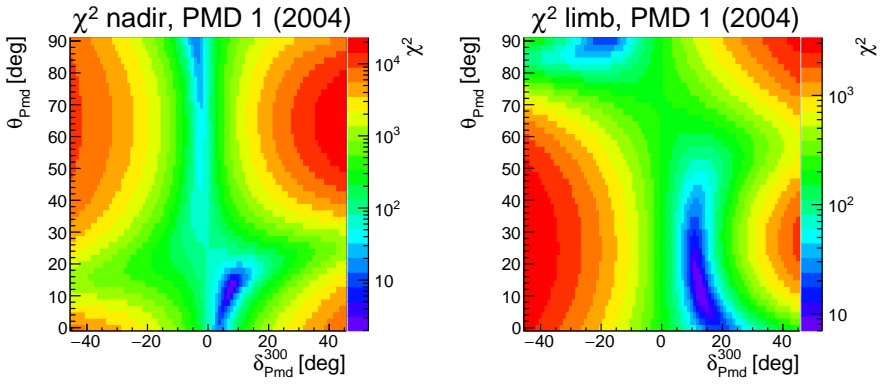

**Figure 13.** $\chi^2$-distribution according to Eq. 30 for PMD 1, nadir (left panel) and limb (right panel) in 2004. The retardance parameter is on the $x$-axis, the retarder angle on the $y$-axis, the color code depicts $\chi^2$ on a logarithmic (rainbow) color scale.

of magnitude. The plot also shows extended regions of minima located distinctly away from the zero point.

Next, the 99.99%-confidence regions for limb and nadir and different detectors (PMDs 1 and 2, Channel 2) are shown in Fig. 14 for each considered year. In 2003, the confidence regions for all detectors and measurement modes overlap in a common region of $\delta$ between $5°$ and $10°$ and $\theta$ between $10°$ and $15°$. Except for PMD 1, the nadir regions also include a zero retardance, while all limb data are located away from zero. For the following years it can be observed that the limb and nadir regions slowly

drift apart, almost consistently for all detectors. While for each measurement mode alone there is still an, albeit small, overlap for all detectors, no common overlap exist even for individual detectors after about 2006. There are several possible reasons for this behavior, none of which can be rigorously excluded: the retarder may be an inadequate model to describe the OBM





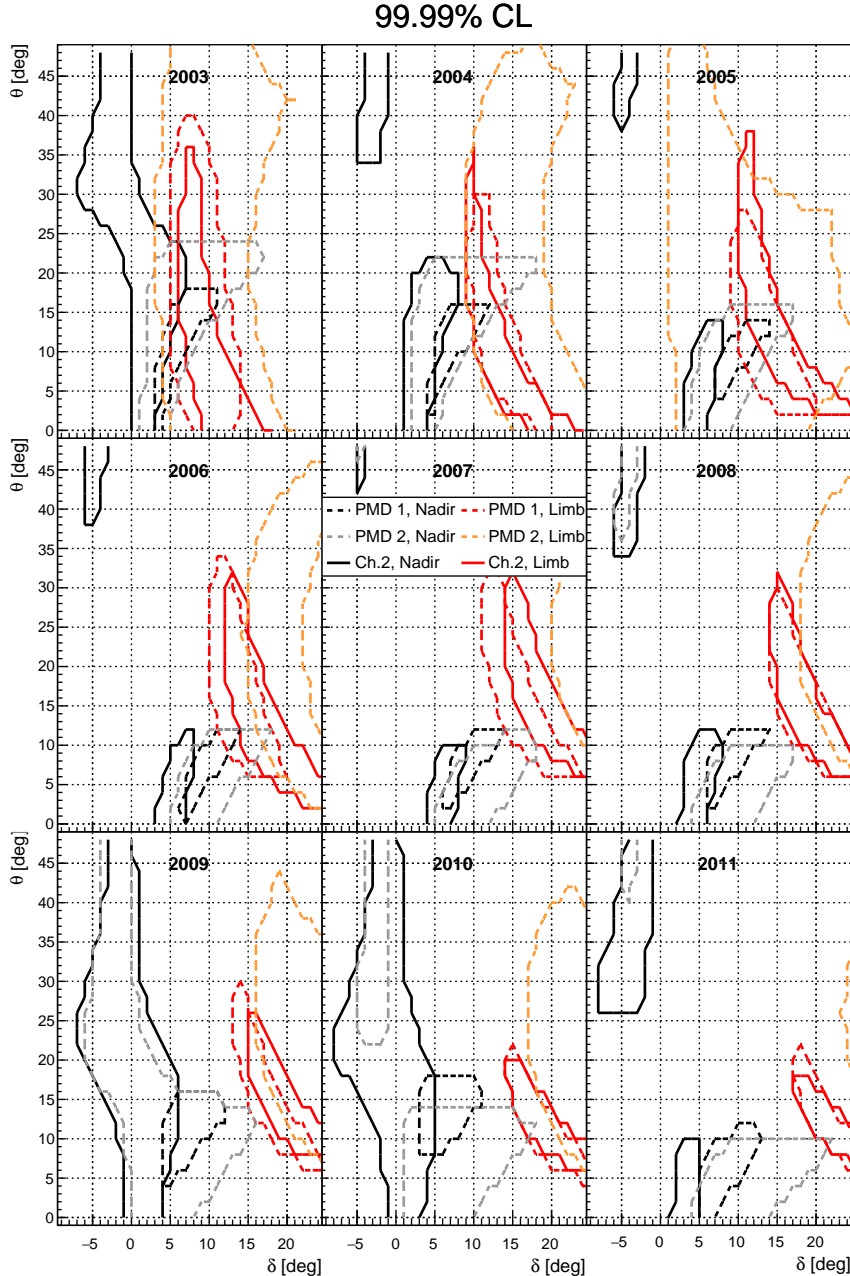

**Figure 14.** The 99.99% confidence region contours for limb and nadir, PMDs 1 and 2 and Channel 2, for each year starting from 2003 (top left) to 2011 (bottom right).





polarization degradation, the mirror degradation parameters may not describe the mirror degradation properly or an unknown, time dependent, systematic error impairs the in-flight measurement and its error estimates. The effect of the mirror model degradation parameters is discussed in section 4.3.

Regarding OBM degradation other than the phase shift it should be considered that, at least for the PMDs, a degradation of
optical elements in the light path after the Brewster reflection inside the prism will not depend on the Stokes vector *before* the prism anymore, therefore it can affect throughput, but not polarization. This argument holds as long as the prism acts as a near perfect polarizer. It may be conceivable that some effect, e.g., radiation damage, changes the refractive index of the prism such that the efficiency of the internal Brewster reflection decreases. In such an event, the refractive index change should be noticed elsewhere as well, for instance in the form of a large wavelength shift of the measured spectra. Such an effect has not been
observed, though.

### 4.2.2 Determination of model parameters and errors

Especially in limb, it is important to have a time dependent in-flight correction to the mirror model MMEs for all wavelengths and PMDs and science channels alike. That is why, even though the observed drift of limb and nadir data does not raise too much confidence in the hypothesis of a time dependent OBM phase shift, the retarder model will be applied to determine the
most likely retarder parameters and their errors as a function of time. This is done by the process marginalizing the respective other parameter, i.e., by integrating each retarder parameter weighted with the likelihood derived from the CDF of the $\chi^2$-distribution (Verde, 2010). The likelihood for any combination of measurement modes and detectors can be obtained from the product of the respective, normalized individual distributions:

$$\mathcal{L}^C\left(\boldsymbol{\Theta}^{Fit}\right) = \prod_i \frac{\mathcal{P}^i(\boldsymbol{\Theta})}{\int \mathcal{P}^i(\boldsymbol{\Theta})d\boldsymbol{\Theta}}, \tag{33}$$

The mean values of a generic fit parameter or a derived parameter $\Theta$ (such as the retarder corrected in flight MMEs) and their errors can be computed from this distribution as follows:

$$\langle\Theta^C\rangle \quad = \quad \int \Theta^C \mathcal{L}^C(\boldsymbol{\Theta})d\boldsymbol{\Theta}, \tag{34}$$

$$\sigma^2(\Theta^C) \quad = \quad \int \left(\Theta - \langle\Theta^C\rangle\right)^2 \mathcal{L}^C(\boldsymbol{\Theta})d\boldsymbol{\Theta} \tag{35}$$

for all considered combinations $C$.

After deriving the retarder parameters in the initial analysis step, an iteration of the entire chain of fits has to be performed. The reflectances and polarization signals are corrected for the polarization of the science channels with the MMEs computed with the retarder parameters from the initial step, using Eq. 15 with the updated MMEs. The effect of this iteration on the value for the retardance is almost negligible, but the retarder angle is shifted down by about 3 degrees.

### 4.2.3 Results

Figure 15 shows the results for the retarder parameters for different observation modes and detector combinations after the iteration step. It demonstrates how adding information from different detectors, e.g., PMD 1 and Channel 2, or combining





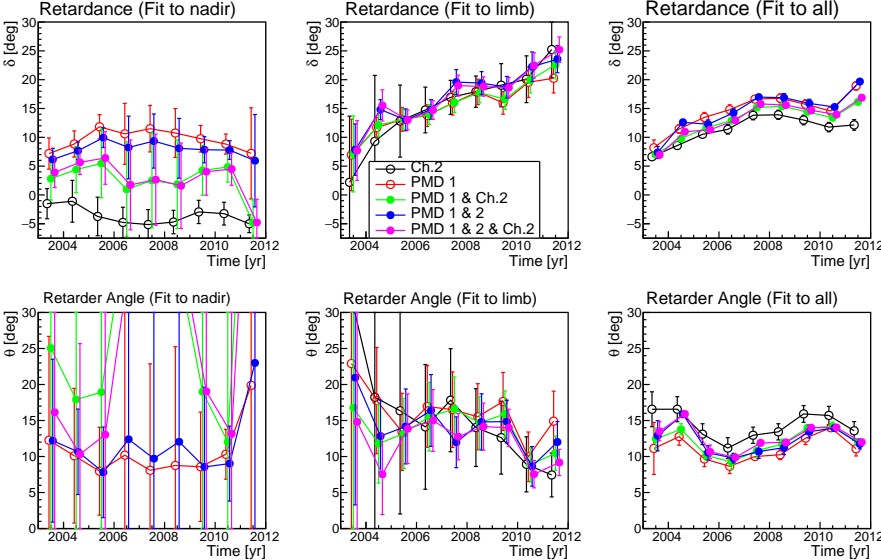

**Figure 15.** Fit results for the parameters $\delta$ (top) and $\theta$ (bottom) vs. time using only nadir (left), only limb (*LUT-Method* for the PMDs, center) or limb and nadir combined (right). Different colors are for different combinations of detectors.

limb and nadir, subsequently changes retarder values and reduces errors. For instance, in nadir there is not much information on the retarder angle because for small retardances, $\chi^2$ (or likewise the probability) does not depend on the retarder angle. In limb, larger retardances are favored and the retarder angle is therefore constrained better. The evident offset between Channel 2 and PMD 1 comes from the peculiar confidence region of Channel 2 which includes large values of $\theta > 45°$ and negative

retardances. Only the combination with the PMDs restricts this region to $\theta < 30°$ and positive retardances. The plot also shows how the combination of limb and nadir data results in a "compromise" which is dominated by the information contained in the limb data. For PMD 2, the sensitivity to the phase shift is considerably smaller than for PMD 1 and Channel 2 due to its wavelength dependence, while the sensitivity to errors in the calibration is larger, as can be seen for instance in the error bars of Fig. 11. Therefore, the final retarder values to be used for polarization determination and correction and correction will

eventually be derived from a combination of limb and nadir, PMD 1 and Channel 2. The values shown in Fig. 15 were obtained with the *LUT-Method* for the PMDs in limb, and *Extrapolation-Method* otherwise. Retarder parameters obtained by using the *Extrapolation-Method* in limb entirely differ by about 1 to maximum 2 degrees for the final result. Parameters change within a similar margin when the off-diagonal elements in the systematic covariance matrix are set to zero.

     The end-to-end MMEs calculated from the retarder model with the fitted parameters are shown in Figs. 16 and 17 together

with the MMEs calculated from the original mirror model and the in-flight data. The errors on the end-to-end MMEs (Eq. 34) are typically less than a few $10^{-3}$ and therefore much smaller than the error on the data, which is why they are not shown on the plot. The retarder model seems to fix the largest differences in nadir, and mostly also in limb. An issue remains with the time dependence in limb, especially towards the later part of the mission. The main effect of the retarder model applied to the



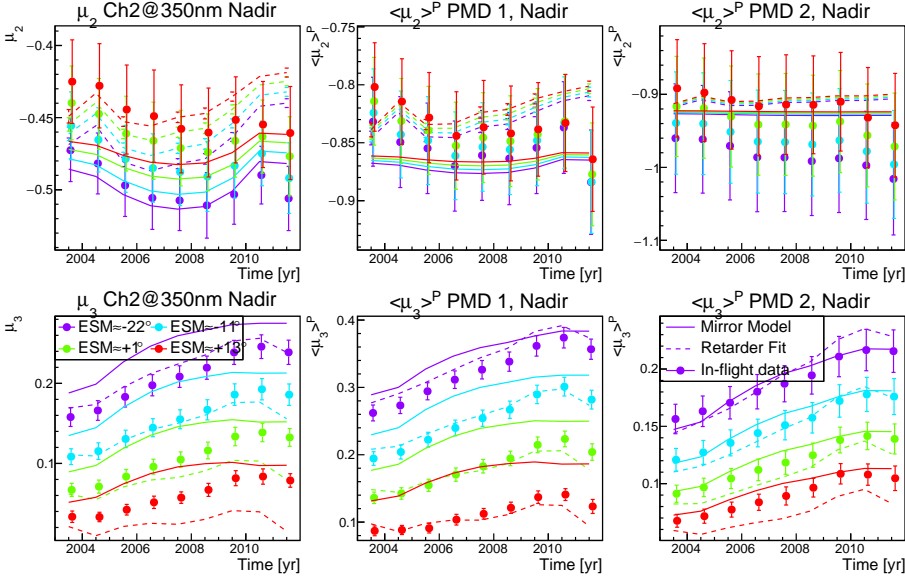

**Figure 16.** Time dependence of in-flight end-to-end Mueller matrix elements $\mu_2$ (top) and $\mu_3$ (bottom) for nadir for four selected ESM angles from measurements (data points) and the scan mirror model without retarder (solid lines), and the mirror model with retarder (dashed lines). The error bars of the data include statistical and total systematic errors. Left: Channel 2 feature, middle: PMD 1, right: PMD 2.The color code signifies the ESM scan angle bin (-22°, -11°, +1°, +13°), increasing from magenta over light blue and green to red.

data will be an improvement of the polarization determination, reducing systematic errors in the limb UV down to about 10% and therefore a factor of about 2. The polarization correction in the UV in Channel 1 and for Channel 2 around 350 nm will also considerably improve. The calibration for the unpolarized response will be nearly unaffected, though.

### 4.3 Dependence of retarder parameters on the scan mirror model

The fit procedure described in section 4.2 can be applied in an analogous way to fit parameters of the mirror degradation, i.e., thicknesses on ESM and ASM mirrors and the refractive index at 350 nm (using PMD 1 and Channel 2) and 480 nm (using PMD 2). The measurement vectors used are the same as for the retarder fit. The combination of different detectors and observation modes is more complex in such a fit because of the relationship of the fit parameters to certain subsets of data. The fit aims at constraining these six independent parameters from selected detector and observation mode combinations which

maximize the information content. The investigated range was from 0 to 100 nm in contaminant thicknesses, from 1 to 1.5 for the real part and from -0.5 to 0 in the imaginary part of the refractive index.

   Together with the unpolarized response constrained to lie within limits consistent with the solar calibration data, the allowed parameter range can be considerably reduced. Fit parameters, for instance thickness and refractive index, tend to be highly anticorrelated, though. This can be explained by the equations describing the effect of a thin layer such as the contaminant on

the mirror substrate, where the relevant term for the reflection coefficient enters as a product of layer thickness and refractive





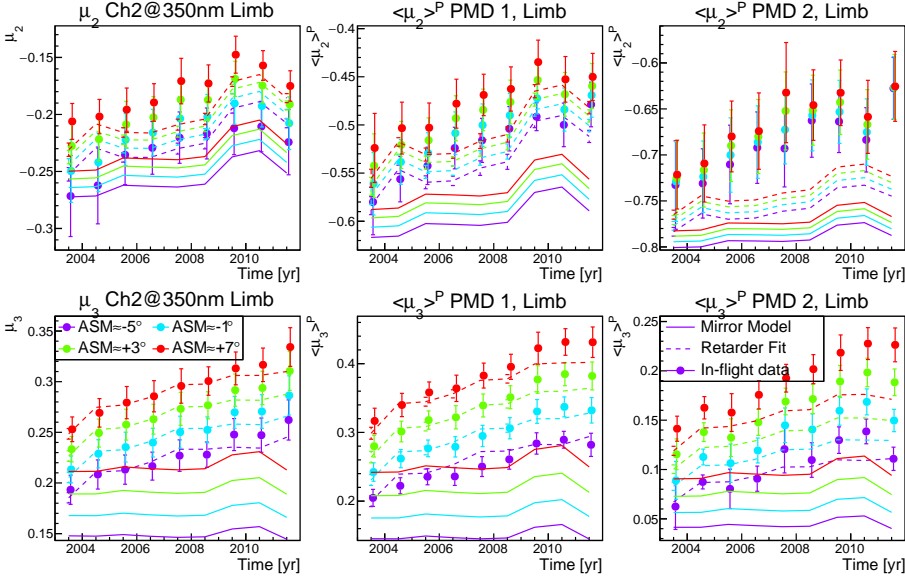

**Figure 17.** Time dependence of in-flight end-to-end Mueller matrix elements $\mu_2$ (top) and $\mu_3$ (bottom) for limb at $TH \approx 28$ km for four selected ASM angles from measurements (data points) and the scan mirror model without retarder (solid lines), and the mirror model with retarder (dashed lines). The error bars include statistical and total systematic errors. Left: Channel 2 feature, middle: PMD 1, right: PMD 2. 1The color code signifies the ASM scan angle bin (-5°, -1°, +3°, +7°), increasing from magenta over light blue and green to red.

index (Krijger et al., 2014). This high correlation inhibits the unambiguous determination of all parameters, but a few qualitative statements can be made based on this investigation.

First, within the scanned parameter range, it is not possible to find a physically reasonable set of parameters that could describe the measured in-flight MMEs without an additional phase shift introduced by the retarder. This finding proves that there is indeed an on-ground to in-flight change inside the OBM. Second, it is possible to find physically reasonable sets of parameters that fit the data when fixing the retarder parameters to those obtained for 2003. That means that a single, on-ground to in-flight change in the OBM would suffice to describe the measurements at all times in combination with mirror model parameters that tend to be different from the ones derived from unpolarized calibration data (see Fig. 10). In particular, the range of possible combinations of thickness and refractive index is inconsistent with those values. In fact, applying a time dependent refractive index with a constant phase shift consistently leads to better agreement between model and data than a time dependent phase shift with a constant refractive index.

At the very least this means that the retarder parameters depend on the mirror model parameters, the more so the thicker the contamination layer is. Furthermore, the findings suggest that the inclusion of polarization data into the fits of the mirror degradation parameters may improve or even resolve the degeneracy between different fit parameters.



### 4.4 OBM vector fits

Nadir and limb data can also be combined to fit the three polarization components of the OBM vector directly for each detector separately. The fit model $\mathbf{X}(\boldsymbol{\Theta})$ (Eq. 30) is linear in this case, with its independent parameters consisting only of the components of the OBM Stokes vector. In contrast to the retarder model, which preserves polarization, in this fit the total OBM

polarization, $p = \sqrt{\mu_2^2 + \mu_3^2 + \mu_4^2}$, can change. The only constraint imposed is that it stays within physical limits, i.e., $p \le 1$.

The results of this test show that the fitted OBM vector $\boldsymbol{\mu}_{OBM}^{Fit}$ is consistent with the phase shifted on-ground vector when scaled with a factor accounting for the change in polarization, i.e.,

$$\boldsymbol{\mu}_{OBM}^{Fit} \approx \boldsymbol{\mu}_{OBM} \mathbf{R}(\delta^{Fit}, \theta^{Fit}) \frac{p^{Fit}}{p^{OBM}}, \tag{36}$$

where $(\delta^{Fit}, \theta^{Fit})$ are the retarder fit parameters from Sect. 4.2.3. This holds in particular for PMDs 1 and 2, which is remark-

able because it also proves consistency with the wavelength dependence ascribed to the stress-induced retardance, Eq. 29. The polarization scale factor only differs from 1 by at most 2%. Also for this fit, a considerable dependence on the mirror model parameters was found, though.

### 4.5 On-ground polarization phase shift

Finally, the use of the combined scan mirror and retarder model enables the interpretation of the on-ground measurements

in terms of an on-ground OBM phase shift, analogous to the analyses of Snel (1999) and Slijkhuis and Frerick (1999). This analysis is based on the presumption that, by original design, the PMDs should have almost 100% sensitivity to perpendicular polarized light entering the OBM, and no sensitivity to $45°$- or circular polarization. This means that

$$\boldsymbol{\mu}_{OBM}^{on-grd} \equiv (1, -p^{OBM}, 0, 0), \tag{37}$$

allowing for a non-perfect polarizer with $p^{OBM} \le 1$. By inverting

$$\boldsymbol{\mu}_{OBM} = \boldsymbol{\mu}_{OBM}^{on-grd} \mathbf{R}(\delta^{on-grd}, \theta^{on-grd}) \tag{38}$$

one can find a set of parameters $(\delta^{on-grd}, \theta^{on-grd})$ that matches the measured on-ground OBM vector $\boldsymbol{\mu}_{OBM}$ within errors. Likewise, by replacing $\boldsymbol{\mu}_{OBM}$ with the result of the in-flight fit $\boldsymbol{\mu}_{OBM}^{Fit}$ (Eq. 36), another set $(\delta^{in-flight}, \theta^{in-flight})$ can be found that relates the non-phase shifted on-ground vector $\boldsymbol{\mu}_{OBM}^{on-grd}$ directly to the fitted in-flight vector, and thus puts the two parameter sets $(\delta^{on-grd}, \theta^{on-grd})$ and $(\delta^{in-flight}, \theta^{in-flight})$ in perspective. The (approximate) results of this exercise are

$\delta^{on-grd} = 35.5° \pm 0.5°,$     $\theta^{on-grd} = 45.0° \pm 2.0°.$

$\delta^{in-flight} = 42.0° \pm 1.5°,$     $\theta^{in-flight} = 35.0° \pm 1.0°.$

Interpreting the on-ground phase shift in terms of stress birefringence $B$ using eq. 26 leads to

$$B \equiv (n_e - n_o)^{on-grd} = 2 \cdot 10^{-6} = 20 \, \text{nm} \cdot \text{cm}^{-1}, \tag{39}$$





a value which is at least 4 times as high as what could be expected from residual birefringence of the material (Keller, 2001). Attributing the entire birefringence to stress, using $R(\lambda_0 = 633 \, \text{nm}) = 35 \, \text{nm} \cdot \text{cm}^{-1} \cdot \text{MPa}^{-1}$ ((Priestley, 2001)) and Eq. 28, an estimate for the on-ground stress amounts to

$$\Delta\sigma = \sigma_1 - \sigma_2 = \frac{n_e - n_o}{R(\lambda = 300 \, \text{nm})} \approx \frac{2 \cdot 10^{-6}}{39.8 \cdot 10^{-7}} \, \text{MPa} = 0.5 \, \text{MPa}, \tag{40}$$

which seems to be quite large considering that the prism mount was originally designed to be stress free. The source of the stress causing the observed birefringence was never found, despite careful investigations of the instrument during on-ground testing. Both a bread board model and the proto-flight model of the instrument showed no abnormal polarization effects at room temperature, and a linear build-up of stress birefringence as temperature was lowered to operational conditions (Snel, 1999). The effect could be reproduced by physically stressing the predisperser prism on component level, but the stress-free mount of the prism prevented this from happening when prism and mount were tested together.

Given the very small thermal expansion coefficient of fused silica and the low intrinsic birefringence, the observed large sensitivity to temperature of the order of $1° \, K^{-1}$ is unlikely to be generated inside the prism itself, but possibly rather inside the instrument or even in the prism mount. Another curious feature is the on-ground retarder angle, which happens to be *the* singular angle that ensures that $\mu_3 \equiv 0$ over the entire wavelength range and all retardances. It seems a bit peculiar that an unintended stress should select exactly this particular direction.

The in-flight phase shift change on the other hand seems to be relatively small compared to the already large on-ground phase shift, and amounts to an approximately 15% difference in stress, with a $10°$ rotation of stress angle. Since the OBM temperature was stable in-flight and did not change with respect to on-ground calibration measurements, the difference in gravitational force has always been suspected to cause the in-flight change. On the other hand, it was also discussed that the entrance window into the vacuum chamber used during the on-ground campaign may at least be responsible for part of the observed phase shift. If that were the case, the on-ground to in-flight change may be simply due to the absence of this so called "OPTEC-window" in flight. With the currently available data, it is not possible to clearly identify the reason for the observed behavior, on-ground and in-flight. Further analysis of all available calibration data may be necessary.

## 5  Conclusions

A novel statistical approach for the in-flight polarization calibration of the SCIAMACHY PMDs and a part of its science channels is presented. It exploits the relationship between polarization and measured reflectance. This approach can in principle be further refined and adapted for the polarization calibration of other instruments which measure polarization, such as GOME, GOME-2 or even POLDER/PARASOL. The results for the channels below $\sim 500 \, \text{nm}$ prove to be stable enough for further investigation of the instrument degradation. First, it was found that the model used to calibrate the scan angle dependent degradation of SCIAMACHY due to contamination of its scan mirrors does neither reproduce the in-flight polarization sensitivities in these channels at the beginning of the mission, nor their time dependence. The effect is largest for the limb data. Second, the observed discrepancies between the model and data in the early part of the mission could be clearly traced back to an on-ground



to in-flight change of the polarization sensitivity inside the optical bench of the instrument. The values are consistent with an extended model that includes both the mirror degradation and stress-induced birefringence inside the predisperser prism. The model was used to fit the parameters of a linear retarder that generates a polarization phase shift.

Several tests were performed to investigate the reason for the observed phase shift, both on-ground and in-flight, as well as its time dependence. The time dependence may indeed be an artifact of the scan mirror model, resulting from the imposed assumptions on the properties of the mirror contaminant. Since the polarization data show a sizeable sensitivity to these properties, the inclusion of these data in addition to the solar measurements in the calibration approach of the scan mirror model may actually help to disentangle some of the ambiguities between different degradation parameters. A retrospect analysis on the on-ground calibration data shows that an unexplained, large phase shift was already present at that time, and that the difference between the on-ground and in-flight phase shifts are comparatively small.

Even though the search for the origin of the phase shift remained inconclusive, the results of this investigation considerably improve the polarization calibration of the measured SCIAMACHY radiances, in particular in limb. As an added benefit, the new calibration, together with an improved algorithm to be applied with version 9 of the operational Level 0 to 1 processor, enables the determination of the atmospheric polarization, i.e., Stokes $q$ for all PMD-wavelengths and $u$ at 850 nm, with a relative accuracy in the order of 10%. Although this is far from the accuracies achieved by dedicated instruments such as POLDER, the new SCIAMACHY polarization data may provide valuable additional information – on a statistical basis – on, e.g, cloud- or aerosol properties for suitable observing geometries.

*Acknowledgements.* SCIAMACHY is a national contribution to the ESA Envisat project, funded by Germany, the Netherlands and Belgium. SCIAMACHY data have been provided by ESA. This work has been funded by DLR Space Agency (Germany), the SCIAMACHY Quality Working Group project (ESA) and by the University of Bremen.



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



**Table 1.** SCIAMACHY PMDs, their spectral range, average wavelength, reference wavelength, corresponding SCIAMACHY science channel, spectrally averaged OBM Mueller matrix elements for the PMDs.

| PMD | Range (nm) | $\langle\lambda\rangle$ (nm) | $\lambda_C$ | Chan. | $\langle M_1\rangle^{PD}$ | $\langle\mu_2\rangle$ | $\langle\mu_3\rangle$ | $\langle\mu_4\rangle$ |
|---|---|---|---|---|---|---|---|---|
| 1 | 305–385 | 352 | 370 | 2 | 0.17 | -0.86 | -0.004 | -0.48 |
| 2 | 430–550 | 484 | 480 | 3 | 0.02 | -0.92 | 0.013 | -0.30 |
| 3 | 590–720 | 656 | 655 | 4 | 0.016 | -0.94 | 0.013 | -0.20 |
| 4 | 780–920 | 852 | 850 | 5 | 0.035 | -0.95 | 0.018 | -0.11 |
| 5 | 1400–1750 | 1570 | 1555 | 6 | 0.099 | -0.97 | 0.005 | 00.11 |
| 7 | 780–920 | 854 | 850 | 5 | 0.07 | 0.10 | 0.94 | 0.33 |