# Peer review of "In-Flight Calibration of SCIAMACHY's Polarization Sensitivity"

_Atmospheric Measurement Techniques, 2017_

## Referee Comment (RC1) · R. Lang (Referee) · 14 Sep 2017

The paper by Liebing et al, investigates the capability of determining the in-flight evolution of the polarisation sensitivity of SCIAMACHY using two empirical model and one rigorous instrument system modelling approach. This is a quite extensive study and readers with less experience in satellite instrument calibration - and especially of those instruments which are sensitive to polarization - may find it difficult to digest. Here is my own, though limited, attempt to summarize the many detailed findings of the paper.

In order to address the observed deficiencies in characterising the in-flight instrument end-to-end response (here expressed in terms of Mueller Matrix elements, MMEs) of polarisation sensitive grating spectrometers like SCIAMACHY, the authors start off by

evaluating two different strategies of using an empirical modelling approach for the SCIAMACHY nadir and limb scanner plus the optical bench module (OBM) common to both based on the observation of "depolarisation" (difference between the expected maximum degree of polarisation under a observed geometry and the measured one). This approach is used in order to establish an empirical relationship between observation geometry and the degree of polarization (involving the observation and/or derivation of q=Q/I and u=Q/I Stokes fractional parameters) based on the large SCIAMACHY measurement data-sets.

The first approach used for the nadir scan-system uses a RTM (Sciatran) to pre-calculate the maximum polarisation for a range of geometries covering all observation types and expresses the observed deviation from the modelled maximum values in a three-parameters empirical functional form for "depolarization". Using this functional form the maximum polarisation can be evaluated for all geometries by a fit to the real observations which are usual all depolarized to some extend (avoiding glint, glory and rainbow conditions). This establishes a relationship between observation geometries and maximum expected q,u Stokes values.

The second approach for the limb scan-system is a LUT relationship established between the degree of polarisation and the ratio of the real signal to the RTM for a given geometry and given q and u values provided by the RTM. This is established using the large limb data-set from SIAMACHY.

Both approaches provide q,u values as they should be observed by the instruments broad-band Polarisation Measurement Devices (PMDs) under (maximum) polarisation conditions and therefore allow the calibration of mu_P (the polarisation sensitivity party of the end-to-end MME of the instrument) values for PMDs, however not for the high spectral resolution main science channels (subscript D). For the latter the authors use a famous SCIAMACHY 350 nm polarisation feature, which is a hot spot of polarisation sensitivity of the grating for channel 2. For this feature a relationship of polarisation signal is established by using the ratio of observed signals to RTM ratios at 350 nm

and outside the feature at 370 nm, which allows to derive the expected mu_d values for science channels at 350 and 370 nm. The observation (san angle) dependence of both mu_P and _D is then parameterized in turn in a polynomial of second order.

In this way end-to-end MMEs for polarisation sensitivity can be established for the limb and the nadir scan-mirror systems plus the OBM, however limited to SCIAMACHY PMD 1 and 2 because of decreasing performance of the approach at higher PMD wavelength and to the 350 nm region of the main science channels.

The paper now moves on to compare these empirical model results to results from a scan-mirror optical model based on first optical principles as previously established by Krijger et al 2014 for SCIAMACHY. The latter is used together with the on-ground OBM MMEs to provide model output over the full SCIAMACHY lifetime for the end-to-end MMEs for PMD 1, 2 and the main channel at 350 nm.

For the nadir scan-mirror system the model reproduces mu2 and mu3 for the three cases quite well when compared to the empirical model results, with a tendency of the model to overestimate mu3 (which is generally increasing in time until approx. 2008) and a tendency of the model to underestimate the observed decrease of mu2 over the same time period, especially for the main science channel 2.

For the limb scan-mirror system the model deviations from the empirical end-to-end MME representation is larger than for the nadir case, with an overestimation of mu2 and an underestimation of mu3 for the whole time period.

The observed deviation are referred to a potential on-ground to in-orbit change of the OBM MME, with a focus on induced stress on the pre-disperser prism during launch, which indeed is usually observed to have changed with respect to on-ground, as observed in a shift of the main science channel overlap points with respect to what was measured on-ground for both the GOME/GOME-2 and SCIAMACHY instruments. Since stress on the pre-disperser prism already was expected to be critical for the instrument performance, the effect of induced stress on this optical component has been

modelled before and is expected to result, apart from a shift in the channel overlap regions, in a phase shift inside the prism.

The authors therefore introduce a retarder model in addition to the OBM and scan mirror MMEs, which models a phase shift based on induced stress, and this retarder model matrix should apply in the same way to the limb and nadir scan-mirror system, i.e. it should improve both model results with respect to the empirical results at the same time. Therefor the authors attempt to minimize the difference between model and empirical model output by fitting the retardance and retarder angle assuming Gaussian distributed errors on the involved MMEs (which for end-to-end MME key-data is generally unrealistic, although attempts have been made in the past to provide such errors using Monte-Carlo simulations). It however turns out that only for the beginning years the retarder model can explain the deviations of the end-to-end MME in a similar way and at the same confidence level for both limb and nadir. Nevertheless the fitted retradance and retarder angle significantly improve the comparison between model and empirical model results, although the induced stress on the pre-disperser seems unrealistically high. Even though the physical interpretation of the retarder model results point at a correlation with a different mechanism not explicitly described, an important result of the study is that this effect seems to be stable over time, as the retarder fit derived from 2003 data improves the results everywhere. So the difference between model and empirical model indeed points at a "one off" on-ground to in-orbit change of the OBM MMEs.

—

The paper is generally very well written, although some introductions to equations and models could be clearer at various points (see detailed comments). The two empirical models established for the limb and the nadir scanner- and OBM system provide a very important extension to the existing tools for polarisation data quality monitoring (like e.g. the well-established "special geometry" method used for near-real time Stokes fraction correction for GOME-2). They will be especially useful for any future

mission data reprocessing from the GOME, GOME-2 and SCIAMACHY instrument types. The paper also demonstrates that the scan mirror model by Krijger et al can be successfully applied to test and compare the end-to-end MMEs and may even be adjusted to account for on-ground to in-flight OBM MME changes (like the to be expected pre-disperser prism stress during launch).

I am generally missing a discussion of the applicability of the empirical approach to other wavelength providing the appropriate measurements would exist. One can see that empirical representations of mu_P can be derived for e.g. a GOME-2 instrument with many more accurate PMD measurements at different wavelength and for both 90 degree polarisation directions. But the 350 nm feature is an unwanted case for SCIAMACHY and the polarisation sensitivity of the rest of the large spectral range is therefore completely unobserved. The users for other instrument cases would therefore need to rely on the validity of the model.

I recommend the manuscript for publication in AMT providing the following detailed but minor comments and editorial and a few lines in the discussion on the applicability of the end-to-end model for other wavelength regions are addressed.

Detailed comments:

p.3, l.1: "The transformation ...". The transformation of what... the OBM model? Or the scanner on-ground model?

p.4, l.18: "The Mueller matrix needs...". The sentence is not very clear... better: "... and if necessary, the reference frames defined...".

p.5, l.15: "Realistically, ...". Sentence and comma needs to be checked. though? ... -> of the?

Section 2.3 2nd paragraph. The orientation of the slit for the ESM configuration should be mentioned. Along flight direction and also parallel to q=-1?

p.8/9, l5ff, Eq 6 to 8: The derivation of Equation 8 is confusing since you want to

determine the ratio but you start with the absolute signal. Suggestion "In the following we determine the polarization from the ratio.... by first determining the signal measured by the PMD S_P as...." Then in line 12 you need to add that you now calculate a ratio P of the polarized signal to the unpolarized signal, which you define as the virtual sum S_D. "By additionally (!) assuming that the ..."

Eq 6.: Why is mu2 and mu3 for P also detector pixel dependend? Isn't the sum over i not only referring to science channel detector pixels covering one PMD measurement?

p.8, l.11: Better mu_ni instead of m_i

p.8, l.7: The -> the

p.8, l15. C1B is not explained where and how it is applied in the previous equations so far.

p.11, Eq 15 and previous paragraph: What you are trying to say here is not very clear. I guess what meant is that every measurement R with R<=R_RTM is corrected with cPRTM derived at the limit R=R_RTM, for which cpRTM = Eq15.

---

## Referee Comment (RC2) · Anonymous Referee #2 · 6 Oct 2017

This is an extensive and detailed study on the inflight polarization calibration of SCIA-MACHY. A statistical analysis of inflight observations is used with radiative transfer model simulations to diagnose changes in the polarization calibration from pre-flight observations and over course of the Envisat mission. These changes are identified as being consistent with various instrument components through a detailed analysis. The results of the study are to be implemented in the operational version of the level 0 to 1 processor and represent a significant improvement over previous calibrations.

The paper is very well written and although long and somewhat tedious in nature, the formal development of this substantive body of work will be quite helpful for future instrument characterization and modelling work. The equations and nomenclature are clear and the figures are well done. In my opinion, other than the one point detailed

below, there are no real outstanding issues with the paper and I recommend publication in AMT.

The vector radiative transfer model is used to calculate maximum possible polarization values for the analysis with several assumed atmospheric state terms and boundary conditions. The approach uses the model to derive limiting values for the polarization of the nadir and limb radiances and is a good idea, in my opinion. The question that arises is about the accuracy of the model, both with regard to the assumed states and boundary conditions as well as the algorithm itself (for example, overestimation of multiple scattering in a plane parallel atmosphere as pointed out by the authors). The model reference paper, Rozanov et al., 2014, shows relatively large differences between SCIATRAN and other vector RT codes for limb radiances, especially in certain geometries. This should at least be mentioned in this paper and if possible the potential impact on the results quantified.

---

## Author Comment (AC1) · 2 Nov 2017

[a4paper,10pt, english]article amssymb amsmath

[Figure]

**AMT 2017-175: Author's Response**

November 2, 2017

We thank Ruediger Lang very much for the thorough and thoughtful review. We will consider the comments in the revised version of the paper. The issue of general applicability of the new calibration approach will be discussed in the Conclusion section of the paper. See detailed answers below.

*"The paper is generally very well written, although some introductions to equations and models could be clearer at various points (see detailed comments). The two empirical models established for the limb and the nadir scanner- and OBM system provide a very important extension to the existing tools for polarisation data quality monitoring (like e.g. the well-established special geometry method used for near-real time Stokes fraction correction for GOME-2). They will be especially useful for any future mission data reprocessing from the GOME, GOME-2 and SCIAMACHY instrument types. The paper also demonstrates that the scan mirror model by Krijger et al can be successfully applied to test and compare the end-to-end MMEs and may even be adousted to account for on-ground to in-flight OBM MME changes (like the to be expected pre-disperser prism stress during launch). I am generally missing a discussion of the applicability of the empirical approach to other wavelength providing the appropriate measurements would exist. One can see that empirical representations of $mu_P$ can*

[Figure]

*be derived for e.g. a GOME-2 instrument with many more accurate PMD measurements at different wavelength and for both 90 degree polarization directions. But the 350 nm feature is an unwanted case for SCIAMACHY and the polarization sensitivity of the rest of the large spectral range is therefore completely unobserved. The users for other instrument cases would there- fore need to rely on the validity of the model. "*

The general applicability to other instruments has not been discussed in much detail because it indeed depends on the specific instrument, its measurement modes and available data. In each case the method would have to be adapted to the available measurements and the information to be inferred. Note that the calibration approach presented here consists of two rather independent parts: The first is the determination of more or less "effective" polarization sensitivities for a limited set of detectors and a rather small wavelength range. The second is the derivation of instrument parameters which reproduce those polarization sensitivities and predict the instrument behavior for an extended wavelength range and other detectors. In the case of SCIAMACHY the combined model for the scan mirrors and the retarder serves such a purpose very well. A similar model could in principle be applied for all GOME-like instruments, though the lack of limb data may decrease the sensitivity to retarder parameters. This could be compensated to some degree by the larger scan angle range.

The overarching requirements for the polarization sensitivity measurement to work are a sufficiently accurate model (or data) applicable to the particular measurement conditions or sufficient statistics and leverage to extrapolate to a limiting model to determine polarization signals, and sufficient coverage in the (q,u)-plane for the determination of polarization sensitivities. Limiting models can be in principle improved by including information on known wind speeds or surface reflectance over particular sites. An approach of calibration over natural targets, as performed for instruments such as MODIS, MERIS, PARASOL etc. may also be considered. This approach has been tested for SCIAMACHY as well, though it turned out that using a few selected oceanic sites gives too small leverage in (q,u) to retrieve meaningful information.

Another option would be to cross calibrate with other instruments such as PARASOL. This had been attempted for SCIAMACHY nadir data using a statistical distribution of polarization vs. reflectance derived from PARASOL data for the same viewing geometries as SCIAMACHY's. The resulting polarization sensitivities for the PMDs were consistent with those from the model approach. Due to the wide range of viewing geometries available from POLDER/PARASOL data, this should also be viable for GOME instruments. The combination of polarization data from PARASOL and the co-located reflectance measurements from MERIS allowed for the determination of science channel sensitivities at the MERIS wavelengths up to 900 nm, using the SCIAMACHY to MERIS reflectance ratio vs. (q,u) as a polarization signal. The reason why this approach was not followed up on was that there seemed to be indeed a calibration error in the PARASOL data at that time, as shown in the attached figure. This figure shows a two-dimensional histogram of $u$ at 850 nm as measured by PARASOL vs. its single Rayleigh scattering value $u_{SS}$, for viewing geometries corresponding to SCIAMACHY's Easternmost scan position in August 2007, and selected to likely contain sun glint scenes over ocean. The dots are the SCIATRAN model results for the same geometries and varying wind speeds. The distribution is evidently shifted sidewards w.r.t. the model. Such a shift can arise from a "contamination" of about 10% $q$ in the $u$ measurement. At lower wavelengths it is less prominent. Given an accurate calibration of the POLDER/PARASOL data, they would prove to be extremely useful in addition, or complementary to, the use of (limiting) models. GOME-2 for instance might be able to use AVHRR data for reference reflectances.

In the UV, below 300 nm, both nadir and limb (science channel) data can be calibrated using directly the ratio of the measured reflectances to those obtained from model which includes an accurate assumption about the $O_3$-concentration. This has been done for SCIAMACHY as well, resulting in a clear polarization signal and derived sensitivities not too far from expectations given the observed behavior at higher wavelength. The lack of accurate $O_3$ data for the relevant altitudes $> 40$ km, however, inhibited an interpretation in terms of instrumental parameters.

In order to address this question in the paper, we propose to add a corresponding paragraph to the Conclusions section, such that the first 2 paragraphs should now read:

A novel statistical approach for the in-flight polarization calibration of the SCIAMACHY PMDs and a part of its science channels is presented. It exploits the relationship between polarization and measured reflectance. This approach can in principle be further refined and adapted for the polarization calibration of other instruments which measure polarization, such as GOME, GOME-2 or even POLDER/PARASOL. The overarching requirements for the polarization sensitivity measurement are a sufficiently accurate model or data applicable to the particular measurement conditions, or sufficient statistics and leverage to extrapolate to a limiting model to determine polarization signals. The resulting polarization signals should cover a significant portion of the (q,u)-plane to provide enough leverage for the determination of polarization sensitivities.

The general applicability to other instruments has not been discussed in much detail here because it depends on the specific instrument, its measurement modes and available data. Limiting models can be in principle improved by including information on known wind speeds or surface reflectance. An extension of well established calibration methods employing natural targets (see Frouin (2013) for an overview) with the *Extrapolation Method* to include polarization may also be worth considering. For SCIAMACHY, though, using a few selected oceanic sites resulted in too small leverage in $(q, u)$ to retrieve meaningful information, for other instruments such as PARASOL or MODIS with many observations of the same scene from different angles this may not pose a problem. For MODIS, for instance, cross calibration with SeaWIFS data and modeled polarization values has been successfully applied (Meister et al., 2009). The modeled relationship between reflectance and polarization may be replaced by suitable data, such as accurately calibrated data from the POLDER or PARASOL instruments. Co-located reflectance measurements may provide a reference reflectance to determine polarization signals when dedicated polarization measurements are not available.

For instance, the combination of MERIS reflectance data co-located with SCIAMACHY measurements, together with a statistical distribution of $(q, u)$ vs. reflectance derived from PARASOL data matched with SCIAMACHY viewing geometries, resulted in an independent measurement of science channel polarization signals at MERIS wavelengths, e.g., 510, 665 and 885 nm. The derived polarization sensitivities for nadir were roughly consistent with expectation (Liebing et al., 2014). The investigated time span was only one month, August 2007, such that any further interpretation of the results was inhibited by too large uncertainties.

Detailed comments:

*p.3, l.1: The transformation . . .. The transformation of what... the OBM model? Or the scanner on-ground model?*

Yes, that's indeed not clear. I suggest the following change: "The application of the model to limb and nadir measurement configurations ..."

*p.4, l.18: The Mueller matrix needs. . . The sentence is not very clear... better: "... and if necessary, the reference frames defined...".*

I propose the following: "The Mueller matrix needs to be given in the same reference frame as the Stokes vector, and if necessary, reference frames for individual components have to be transformed to this particular reference frame."

*p.5, l.15: Realistically, . . . Sentence and comma needs to be checked. though? ... -of the?*

I've changed the sentence to: "Realistically, the limited information contained in the combination of calibration measurements used only allows for the determination of a single, wavelength dependent refractive index which is constant in time, and the time dependent thicknesses on each of the scan mirrors."

*Section 2.3 2nd paragraph. The orientation of the slit for the ESM configuration should be mentioned. Along flight direction and also parallel to q=-1?*

We propose to add the following at the end of section 2.3:

"In Fig. ??, the scan mirror configurations and viewing geometries for the nadir and limb observation modes are depicted on the left and right, respectively. In the nadir configuration, the instrument slit is oriented along the flight direction and therefore perpendicular to the meridional plane, which lies in the scan direction. In limb, the projection of the slit is along the horizon and thus again perpendicular to the plane connecting the line-of-sight and local zenith or tangent point. Therefore, in both cases, $q = -1$ if the polarization direction (in the atmospheric Stokes frame) is along the instrument slit projection. "

*p.8/9, l5ff, Eq 6 to 8: The derivation of Equation 8 is confusing since you want to determine the ratio but you start with the absolute signal. Suggestion In the following we determine the polarization from the ratio.... by first determining the signal measured by the PMD $S_P$ as.... Then in line 12 you need to add that you now calculate a ratio P of the polarized signal to the unpolarized signal, which you define as the virtual sum $S_D$. By additionally (!) assuming that the ...*

I hope this reshuffling of the entire paragraph makes it a bit clearer:

"The polarization is determined by equating the synchronized and integrated (over the exposure time of the science channel) PMD signal with the calibrated science channel signal, scaled with the PMD response and integrated over the PMD spectral band:

$$S^{\mathsf{P}} = \mu_1^{PD} \cdot \sum_i S_i^{\mathsf{D}} M_{1,i}^{\mathsf{PD}} \frac{1 + \mu_{2,i}^{\mathsf{P}} q + \mu_{3,i}^{\mathsf{P}} u}{1 + \mu_{2,i}^{\mathsf{D}} q + \mu_{3,i}^{\mathsf{D}} u} , \quad \text{with} \tag{1}$$

$$M_{1,i}^{\mathsf{PD}} = \frac{M_{11,i}^{\mathsf{P}}}{M_{11,i}^{\mathsf{D}}} . \tag{2}$$

The sum goes over all pixels in the relevant spectral range, the superscripts $P$ and $D$ indicate PMD and science detectors, respectively. The $\mu_{n,i}^{P,D}$ are end-to-end Mueller

vector elements and vary with observation mode and scan angle. The factor $\mu_1^{PD}$ is an additional in-flight calibration factor that accounts for calibration offsets in the relative PMD to science channel response to unpolarized light. Assuming that the polarization and the polarization sensitivity varies sufficiently slowly with wavelength, this equation can be further simplified:

$$S^{\mathsf{P}} = \mu_1^{PD} \cdot \sum_i S_i^{\mathsf{D}} M_{1,i}^{\mathsf{PD}} \frac{1 + \langle \mu_2^{\mathsf{P}} \rangle q + \langle \mu_3^{\mathsf{P}} \rangle u}{1 + \langle \mu_2^{\mathsf{D}} \rangle q + \langle \mu_3^{\mathsf{D}} \rangle u}. \tag{3}$$

The quantities in the angular brackets are now wavelength independent and refer to the intensity weighted spectral average of the polarization sensitivities:

$$\langle \mu_n^{\mathsf{P,D}} \rangle = \frac{1}{S^D} \sum_i S_i^{\mathsf{D}} M_{1,i}^{\mathsf{PD}} \mu_{ni}^{\mathsf{P,D}}, \quad n = 2, 3 \tag{4}$$

with

$$S^D = \sum_i S_i^{\mathsf{D}} M_{1,i}^{\mathsf{PD}}. \tag{5}$$

The term $S^D$ is also called *virtual sum* and describes the expected PMD signal for zero polarization, given the science channel signal and the relative detector responses. With Eq. 5 a *polarization signal*, $P$, can be defined as the ratio of the PMD signal to the virtual sum,

$$P \equiv \mu_1^{PD} \frac{S^P}{S^D} \approx \frac{1 + \langle \mu_2^{\mathsf{P}} \rangle q + \langle \mu_3^{\mathsf{P}} \rangle u}{1 + \langle \mu_2^{\mathsf{D}} \rangle q + \langle \mu_3^{\mathsf{D}} \rangle u} \tag{6}$$

which depends on polarization only."

*Eq 6.: Why is mu2 and mu3 for P also detector pixel dependent? Isnt the sum over i not only referring to science channel detector pixels covering one PMD measurement?*

Eq. 6 would allow for wavelength dependent PMD sensitivities (which is in principle possible), though in the current version of calibration data they are not. The wavelength dependence of the PMD mu2,3 is retained here for historical reasons, and because technically the calibration data are given as a function of wavelength. Some of the previous versions of calibration data on mu2,3 indeed showed a wavelength dependence.

*p.8, l.11: Better $mu_ni$ instead of $m_i$ p.8, l.7: The $->$ the*

This has been corrected in the manuscript.

*p.8, l15. C1B is not explained where and how it is applied in the previous equations so far.*

C1B has been replaced with the previously defined $\mu_1^{PD}$.

p.11, Eq 15 and previous paragraph: What you are trying to say here is not very clear. I guess what meant is that every measurement R with $R <= R_R TM$ is corrected with cPRTM derived at the limit $R = R_R TM$, for which $cpRTM = Eq15$.

Yes, that's correct. To clarify this, the preceding sentence has been slightly changed:

"Since in this step of the analysis the actual polarization values for each data point data are not yet available, the polarization values used in the correction are the RTM values themselves, i.e., each data point is corrected for the *maximum* polarization $(q_{RTM}, u_{RTM})$ at $R = R_{RTM}$:"

**References**

Frouin, R., ed.: In-flight Calibration of Satellite Ocean-Colour Sensors, vol. No. 14 of *Reports of the International Ocean Colour Coordinating Group*, IOCCG, Dartmouth, Canada, http://www.ioccg.org/reports/IOCCG_Report_14_2013.pdf, 2013.

Liebing, P., Snel, R., Bramstedt, K., and Krijger, M.: An Assessment of the In-flight Polarization Response of SCIAMACHY, AGU Fall Meeting Abstracts, http://www.iup.uni-bremen.de/sciamachy/polarisation/AGU/SciaPol_pix.pdf,

Meister, G., Franz, B. A., Kwiatkowska, E. J., Eplee, R. E., and McClain, C. R.: Detector dependency of MODIS polarization sensitivity derived from on-orbit characterization, vol. 7452, pp. 7452 – 7452 – 12, 10.1117/12.825385, http://dx.doi.org/10.1117/12.825385, 2009.

[Figure]

figure-1.pdf

**Fig. 1.** Parasol data on $u$ vs. its single scattering value for August 2007 over ocean, with viewing geometries selected to match SCIAMACHY's Easternmost scan position and high likelihood for sun glint. The small dots are the result of SCIATRAN simulation with different wind speeds.

---

## Author Comment (AC2) · 2 Nov 2017

[a4paper,10pt, english]article amssymb amsmath

[Figure]

**AMT 2017-175: Author's Response**

November 2, 2017

We thank the referee very much for the positive and concise review. The referee's comment on intrinsic model errors will be discussed below and addressed in the revised version of the paper.

*Comment: "The vector radiative transfer model is used to calculate maximum possible polarization values for the analysis with several assumed atmospheric state terms and boundary conditions. The approach uses the model to derive limiting values for the polarization of the nadir and limb radiances and is a good idea, in my opinion. The question that arises is about the accuracy of the model, both with regard to the assumed states and boundary conditions as well as the algorithm itself (for example, overestimation of multiple scattering in a plane parallel atmosphere as pointed out by the authors). The model reference paper, Rozanov et al., 2014, shows relatively large differences between SCIATRAN and other vector RT codes for limb radiances, especially in certain geometries. This should at least be mentioned in this paper and if possible the potential impact on the results quantified."*

This is a very valid point, and the investigation of the effects of intrinsic model errors on the results, mainly on the polarization sensitivities, represents a large body of the work performed preceeding this publication. The reason why the discussion of these matters

has been kept to a minimum is that with a more detailed discussion, the paper would have been even longer and more tedious. The *Extrapolation method* relies on the assertion that in most cases, a distribution of scenes with varying albedo and aerosol conditions can be extrapolated using their reflectance distribution to a converging point with minimum reflectance and maximum polarization. This has been confirmed with a limited sample of simulated scenarios in nadir, and extensively for limb. Errors of the extrapolation have been minimized through careful choice of the included geometries and locations, as described in section 3.1. They cannot be fully avoided though, and one reason for integrating data over the relatively long period of one year was to add data points from slightly different geometries or seasons into a cell of $(q, u)$ so as to achieve perhaps less systematic errors at the expense of higher noise. However, especially for nadir there is indeed no robust quantitative estimate on the effect of systematic extrapolation errors. Instead, the study concentrated on specific variations of the zero-point in nadir, e.g., by varying the wind speed over ocean or completely neglecting the surface reflectance (pure Rayleigh scattering). Data over land for limited wavelengths and scan angles provided additional consistency checks. The study basically resulted in the specific choice of reliable wavelengths ($< 500$ nm) and wind speeds. The optimal wind speed and its variation were for instance chosen by comparing the resulting distributions of $R/R_{RTM}$ and requiring that most of the data, especially at larger wavelengths, have $R/R_{RTM} \geq 1$. In limb, the results of the extrapolation method can be directly compared to those from the *LUT method* and give an estimate of errors related to extrapolation and assumptions in deriving the LUT. Further studies regarded the potential influence of absorption by trace gases such as $O_3$ and $H_2O$ which was found to be negligble. Additional confidence in the results is given by Fig. 13, which shows that at least for the beginning of the mission a $\chi^2$-distribution consistent with associated systematic errors.

Another issue is the theoretical accuracy of the RTM for a given scenario, both in the limiting cases as well as the distribution for the LUT method. For nadir, comparisons of several RTM for Rayleigh scattering and BRDF modelling showed sufficiently high accuracy (Kokhanovsky et al., 2010; Rozanov et al., 2014). In limb, the approximation of the multiple scattering contribution by a plane parallel atmosphere leads to the afore-mentioned overestimation of reflectances, increasing with TH. Further comparisons between the MYSTIC Monte Carlo model and SCIATRAN showed an overestimation of the depolarization as well Rozanov et al. (2014). For the extrapolation method, and the normalization point of the LUT method, the error is larger at shorter wavelengths due to the contribution of multiple Rayleigh scattering. At the THs considered in this analysis, this error does not exceed 0.01 for both $q$ and $u$, and depends on geometry. At larger wavelengths multiple scattering is introduced by surface reflectance or aerosols and clouds. It therefore does affect the predictions of the *LUT method* at higher values of the reflectance or $R/R_{RTM}$. This method, however, suffers from many more prevailing errors, such as the choice of the scenarios contributing to the averaged polarization vs. reflectance curve as shown in Fig. 5. The effect of the approximation of multiple Rayeigh scattering has in fact been taken into account in the calculation of the limb systematic errors. A set of polarization sensitivities has been obtained by correcting the limiting reflectance and polarization values with a rough parametrization of the observed differences between SCIATRAN and MYSTIC values. The resulting differences in polarization sensitivities lie typically within the error bands generated by the $\delta R/R = \pm 0.05$ variations. It has been added to that error, but does not contribute significantly to it.

We suggest the following changes to the paper to address ths concern:

In section 3.3.2., after the first paragraph, we add:
"For limb data, an estimate of a theoretical model uncertainty arising from the plane parallel approximation of the multiple scattering contribution on both reflectance and polarization is included in the total systematic error. The uncertainty has been esti-mated from a comparison between SCIATRAN and the MYSTIC (Emde and Mayer, 2007) Monte Carlo model (Rozanov et al., 2014). Its contribution to the systematic error of the polarization sensitivities is in general smaller than the error arising from the

normalization uncertainty and is therefore not depicted explicitly."

At the end of that same section, we propose the following statement:

"Generally, studies on the sensitivity of both methods with respect to model input parameters and data selection criteria indicate that at wavelengths above 500 nm the results become rather unstable. At lower wavelengths the measurements are less influenced by the unknown atmospheric and surface conditions, such that results are more reliable. In the following, the discussion is therefore restricted to the results obtained from PMDs 1 and 2 and Channel 2."

**References**

Emde, C. and Mayer, B.: Simulation of solar radiation during a total eclipse: a challenge for radiative transfer, Atmospheric Chemistry and Physics, 7, 2259–2270, 10.5194/acp-7-2259-2007, https://www.atmos-chem-phys.net/7/2259/2007/, 2007.

Rozanov, V., Rozanov, A., Kokhanovsky, A., and Burrows, J.: Radiative transfer through terrestrial atmosphere and ocean: Software package {SCIATRAN}, Journal of Quantitative Spectroscopy and Radiative Transfer, 133, 13 – 71, http://dx.doi.org/10.1016/j.jqsrt.2013.07.004, http://www.sciencedirect.com/science/article/pii/S0022407313002872, 2014.

Kokhanovsky, A. A., Budak, V. P., Cornet, C., Duan, M., Emde, C., Katsev, I. L., Klyukov, D. A., Korkin, S. V., C-Labonnote, L., Mayer, B., Min, Q., Nakajima, T., Ota, Y., Prikhach, A. S., Rozanov, V. V., Yokota, T., and Zege, E. P.: Benchmark results in vector atmospheric radiative transfer, Journal of Quantitative Spectroscopy and Radiative Transfer, 111, 1931 – 1946, https://doi.org/10.1016/j.jqsrt.2010.03.005, http://www.sciencedirect.com/science/article/pii/S0022407310000919, 2010.